# On the agreement between bibliometrics and peer review: Evidence from the Italian research assessment exercises

**Alberto Baccini** [1]*, **Lucio Barabesi**[1], **Giuseppe De Nicolao**[2]

**1** Department of Economics and Statistics, University of Siena, Siena, Italy, **2** Department of Electrical, Computer and Biomedical Engineering, University of Pavia, Pavia, Italy

\* alberto.baccini@unisi.it

**Data Availability Statement:** All data are available as Supplementary information. Thay are also downloadable at 10.5281/zenodo.3727460.

## Abstract

This paper analyzes the concordance between bibliometrics and peer review. It draws evidence from the data of two experiments of the Italian governmental agency for research evaluation. The experiments were performed by the agency for validating the adoption in the Italian research assessment exercises of a dual system of evaluation, where some outputs were evaluated by bibliometrics and others by peer review. The two experiments were based on stratified random samples of journal articles. Each article was scored by bibliometrics and by peer review. The degree of concordance between the two evaluations is then computed. The correct setting of the experiments is defined by developing the design-based estimation of the Cohen's kappa coefficient and some testing procedures for assessing the homogeneity of missing proportions between strata. The results of both experiments show that for each research areas of science, technology, engineering and mathematics the degree of agreement between bibliometrics and peer review is—at most—weak at an individual article level. Thus, the outcome of the experiments does not validate the use of the dual system of evaluation in the Italian research assessments. More in general, the very weak concordance indicates that metrics should not replace peer review at the level of individual article. Hence, the use of the dual system in a research assessment might worsen the quality of information compared to the adoption of peer review only or bibliometrics only.

## 1 Introduction

Efficient implementation of a research assessment exercise is a common challenge for policy makers. Even if attention is limited to scientific quality or scientific impact, there is a trade-off between the quality of information produced by a research assessment and its costs. Until now, two models have prevailed [1]: a first model based on peer review, such as the British Research Excellence Framework (REF), and a second model based on bibliometric indicators, such as Australian Excellence in Research (ERA), for the years preceding 2018. The first model is considered more costly than the second. In the discussion on the pros and cons of the two models, a central topic deals with the agreement between bibliometrics and peer review. Most

**Funding:** Alberto Baccini is the recipient of grants by the Institute For New Economic Thinking Grant Institute For New Economic Thinking Grant ID INO17-00015 and INO19-00023.

**Competing interests:** NO authors have competing interests.

part of the scholarly works has analyzed the REF by adopting a post-assessment perspective [2]. Indeed, results of the REF at various levels of aggregation are compared with those obtained by using bibliometric indicators. A clear statistical evidence on the concordance of bibliometrics and peer review would represent a very strong argument in favor of the substitution of the latter with the former. Indeed, the claim for such a substitution—based on agreement and minor costs—could likely appear pragmatic and hence more acceptable for academics than the argument based on juxtaposition of "objective bibliometric data" and "subjective peer reviews" (among others, see e.g. [3]).

However, there are two problems hindering the adoption of the bibliometric model for research assessment. The first is how to handle the scientific fields for which bibliometrics is not easily applicable, namely social sciences and humanities. The second is how to manage research outputs not covered in bibliographic databases, such as books or articles in national languages. In these cases, no substitution is possible and peer review appears as the unique possible tool for evaluating research outputs.

As a consequence, a third model of research assessment has emerged, where bibliometrics and peer review are jointly adopted: some research outputs are evaluated by bibliometrics and others by peer review. The evaluations produced by the two techniques are subsequently mixed together for computing synthetic indicators at various levels of aggregation. The Italian governmental agency for research evaluation (ANVUR) applied extensively this model in its research assessment exercises (VQR), and called it "dual system of evaluation" [4]. In reference to this model, the question of the agreement between bibliometrics and peer review has a constitutive nature. Indeed, a high agreement would ensure that final results of a research assessment—at each possible level of aggregation—are not biased by the adoption of two different instruments of evaluation. In the simplest scenario, this issue might happen when bibliometrics and peer review produce scores which substantially agree, for instance, when the research outputs evaluated by bibliometrics receive the same score by peer review—except for random errors. In contrast, let us consider a second scenario where scores produced by bibliometrics and peer review do not agree: for instance, bibliometrics produces scores systematically lower or higher than peer review. In this more complex case, the disagreement might not be a problem solely if the two systems of evaluation are distributed homogeneously, e.g. at random, among units of assessment. Even if the concordance is not accurate at the individual article level, the errors may offset at an aggregate level [2, 5]. In sum, the agreement between bibliometrics and peer review is functional for validating results of the assessment.

ANVUR tried to validate the use of the dual system of evaluation by implementing two extensive experiments on the agreement between bibliometrics and peer review, for each national research assessment of the years 2004-2010 (VQR1) and 2011-2014 (VQR2). The two experiments are hereinafter indicated as EXP1 and EXP2, respectively. They consisted in evaluating a random sample of articles by using both bibliometrics and peer review, and, subsequently, in assessing their degree of agreement at an individual publication level. ANVUR presented the results of EXP1 and of EXP2 as the evidence of a substantial concordance between bibliometrics and peer review. In turn, this agreement would validate the use of the dual system of evaluation and the final results of the research assessements.

Two of the authors of the present paper documented the flaws of EXP1 and contested the interpretation of data as indicative of a substantial agreement [6–9]. The present paper takes advantage of the recent availability of the raw data of the two experiments, in order to deepen the analysis and reach conclusive results on issues that had remained open due to the sole availability of aggregated data. Therefore, this paper aims to replicate the ANVUR analysis in order to draw a solid evidence on the concordance between bibliometrics and peer review.

The paper is organized as follows. In Section 2 the literature on the two Italian experiments is framed in the general discussion on the agreement between bibliometrics and peer review. Section 3 presents the structure of EXP1 and EXP2 by reminding the essential issues of the Italian research assessment exercises. Section 4 introduces the main research questions on the sampling design and the measures of agreement. Section 5 develops the correct framework for the design-based estimation of the Cohen's kappa coefficient. Section 6 presents the estimates of Cohen's kappa coefficients for EXP1 and EXP2, by comparing the current results with ANVUR's findings. In Section 7, a further problem with missing data in EXP2 is presented and the homogeneity of missing proportions between scientific areas is assessed. Section 8 discusses results and concludes with some suggestions for research evaluation policy.

## 2 A short review of the literature

Most part of the literature on the agreement between bibliometrics and peer review considers the British REF. Overviews of this literature are provided by [2, 5, 10]. It is therefore possible to limit the discussion to a central issue which is functional to the development of this paper. By and large, results on agreement do not converge when different approaches and statistical tools are used. Notably, the analysis conducted by the Higher Education Funding Council for England (HEFCE) in the so-called *Metric Tide* report "has shown that individual metrics give significantly different outcomes from the REF peer review process, showing that metrics cannot provide a like-for-like replacement for REF peer review" [11]. This analysis was performed at an individual article level by comparing the quality profile attributed by peer reviews to a set of bibliometric indicators for articles submitted to REF. Traag and Waltman [2] criticized results of the *Metric Tide* report by arguing that the individual publication level "is not appropriate in the context of REF". They claimed that the appropriate level is the institutional one, since "the goal of the REF is not to assess the quality of individual publications, but rather to assess 'the quality of research in UK higher education institutions'. Therefore, the question should not be whether the evaluation of individual publications by peer review can be replaced by the evaluation of individual publications by metrics but rather whether the evaluation of institutions by peer review can be replaced by the evaluation of institutions by metric". In a similar vein, Pride and Knoth [5] documented that a high concordance between bibliometric and peer-review indicators for REF is achieved when the analysis is conducted at an institutional level.

These claims should be framed in a "post-assessment" perspective, where the issue at stake is to verify the coherence between results obtained by applying one evaluative technique or the other at the desired institutional level. In the case of REF the coherence to be verified is between the adopted technique, i.e. peer review, and the alternative, i.e. bibliometrics. This viewpoint is very different from that developed in the Italian experiments and considered in this paper. In the present case, the question is whether it is possible to interchangeably use bibliometrics and peer review at an individual article level. To this end, the analysis of the agreement between bibliometrics and peer review at the level of individual publications is therefore fully justified. In turn, Traag and Waltman [2] support the study of the concordance at an individual publication level when the issue is the possibility that bibliometrics replaces peer review at an individual level. In reference to *Metric Tide* report, they explicitly wrote that "the analysis at the level of individual publications is very interesting. The low agreement at the level of individual publications supports the idea that metrics should generally not replace peer review in the evaluation of a single individual publication" [2].

As anticipated, ANVUR implemented EXP1 and EXP2 in order to justify the use of a dual system of evaluation in VQR1 and VQR2. As to EXP1, results were initially published as part

of the official report of the research assessment exercise [12]. In the official report results are synthesized by stating that "there is a more than adequate concordance between evaluation carried out through peer reviews and through bibliometrics. This results fully justifies the choice (...) to use both techniques of assessment" [12 Appendix B, pp. 25-26, translation by the authors] (See also [6]) Ancaiani et al. [4] republished the complete results of EXP1, by claiming a "fundamental agreement" between bibliometrics and peer review "supporting" the choice of using both techniques in the VQR1. Moreover, they also interpreted the experiment as indicating that "combining evaluations obtained with peer review and bibliometric methods can be considered more reliable than the usual practice of combining two or more different evaluations obtained by various reviewers of the same article".

The specific results obtained in EXP1 for the field of Economics and Statistics were largely disseminated. Bertocchi and coauthors published as far as five identical working papers where they interpreted the results of EXP1 by claiming that bibliometrics and peer review "are close substitutes" (among the others [13]). In the version finally published in a scholarly journal, they concluded that "the agencies that run these evaluations could feel confident about using bibliometric evaluations and interpret the results as highly correlated with what they would obtain if they performed informed peer review" [14].

The results and the interpretation of EXP1 were challenged by two of the authors of the present paper on the basis of published data only, since they were unable to access raw data at the time undisclosed by ANVUR (the whole thread of papers, comments and replies includes [6–9, 15, 16]). The first critical appraisal was about the interpretation of the degree of concordance. Baccini and De Nicolao [6, 7] argued that, according to the available statistical guidelines, the degree of concordance between bibliometrics and peer review has to be interpreted as "unacceptable" or "poor" for all the considered research fields. The unique exception—confirmed by a statistical meta-analysis of the data—was Economics and Statistics, for which the protocol of the experiment was substantially modified with respect to the other fields. Baccini and De Nicolao [8, 9] also raised some questions on the sampling protocol used for EXP1, which are considered in details also in this paper.

As for to EXP2, the results were published in the official report [17] and presented in a conference [18]. The synthesis of the results apparently confirmed the outcome of EXP1. The results of EXP2, summarized in the conclusion of the report, state that there is a "non-zero correlation" "between peer review evaluation and bibliometric evaluation". The degree of agreement is "modest but significant. Of particular importance is the result that the degree of concordance (class and inter-row) between the bibliometric evaluation and the peer evaluation is always higher than the one existing between the two individual peer reviews" [17, Appendix B, p. 33, translation by the authors]. These results are interpreted as indicating that "the combined used of bibliometric indicators for citations and journal impact may provide a useful proxy for peer review judgements" [18].

As anticipated, this paper aims to draw definitive evidence from the two experiments. This analysis is possible since ANVUR accepted to disclose the anonymous individual data of both EXP1 and EXP2. The mail to the President of ANVUR containing the request is dated March 12th 2019. The decision of disclosing the data was communicated by mail dated March 26th 2019. Access to the data was open on April 9th 2019. It is therefore possible to replicate the results of EXP1 and EXP2, by verifying in details ANVUR methods and calculations. Replication is solely possible at the research area levels, since—according to a communication dated 16th March 2019—the data for the sub-areas "are no longer available" in the ANVUR archives. For a correct understanding of the research questions, the following section presents a description of EXP1 and EXP2 in the context of the Italian research assessments.

## 3 A brief description of the Italian experiments

EXP1 and EXP2 were designed and performed during VQR1 and VQR2, respectively. Italian research assessement exercises aimed to evaluate research institutions, research areas and fields, both at national and institutional level (i.e. universities and departments). Synthetic indicators were obtained by aggregating the scores received by the research outputs submitted by the institutions. All the researchers with a permanent position had to submit a fixed number—with few exceptions—of research outputs (3 in VQR1 and 2 in VQR2). VQR1 and VQR2 were organized in 16 research area panels. Research areas were distinguished between "bibliometric areas", i.e. science, technology, engineering and mathematics (namely Mathematics and Informatics (Area 1), Physics (Area 2), Chemistry (Area 3), Earth Sciences (Area 4), Biology (Area 5), Medicine (Area 6), Agricultural and Veterinary Sciences (Area 7), Civil Engineering (Area 8b), Industrial and Information Engineering (Area 9)), and "non bibliometric areas", i.e. social science and humanities (namely Architecture (Area 8a) Antiquities, Philology, Literary studies, Art History (Area 10), History, Philosophy, Pedagogy and Psychology (Areas 11a and 11b), Law (Area 12), Economics and Statistics (Area 13), Political and Social Sciences (Area 14)).

Both research assessments performed evaluations of the submitted research outputs by using a "dual system of evaluation" where some outputs were evaluated by bibliometric algorithms and others by "Informed Peer Review" (IPR). Informed peer review indicates that reviewers were asked to evaluate a submitted research item by being provided with its complete metadata and, if available, with its bibliometric indicators. Actually, this dual system of evaluation regarded only the bibliometric areas plus Economics and Statistics (Area 13). Indeed in the non-bibliometric areas, panels evaluated all the submitted research products exclusively by peer review. In the bibliometric areas, instead, while books, book chapters and articles in not-indexed journals were evaluated by IPR, journal articles were evaluated for the most part by applying bibliometric algorithms. VQR1 and VQR2 adopted two different bibliometric algorithms. Both algorithms combined the number of citations received by an article and a journal indicator, e.g. the impact factor. The complete description of the algorithms and their critical appraisal can be found in [6, 19–21]. Both algorithms were built in such a way that, if the two indicators were coherent, they generated a categorical score (B-score) and a corresponding numerical value used for computing aggregate results for institutions. Namely, in the VQR1 there were four categories: Excellent (A, score 1), Good (B, score 0.8), Acceptable (C, score 0.5), Limited (D, score 0); in the VQR2 there were five categories: Excellent (A, score 1), Elevated (B, score 0.7), Fair (C, score 0.4), Acceptable (D, score 0.1), Limited (E, score 0). If the two bibliometric indicators gave incoherent indications for an article, e.g. high number of citations and low impact factor or vice-versa, the algorithm classified it as "IR" (Inconclusive Rating) and it was evaluated by IPR. In both VQR1 and VQR2, Area 13 (Economics and Statistics) did not adopt the bibliometric algorithms for evaluating articles. They were substituted by classifications of journals directly developed by the area panel, including the same number of categories as in the algorithms. Therefore, all the articles received the score of the journal where they were published and no article was classified as IR.

IPR was identically organized in the two research assessments. A publication was assigned to two members of the area panel, who independently chose two anonymous reviewers. The two reviewers performed the IPR of the article by using a predefined format—slightly different between the two research assessments and also between panels in the same assessment. Each referee assigned a final evaluation according to the same final categories adopted for bibliometrics. These final evaluations are conventionally indicated as P1-score and P2-score. Then,

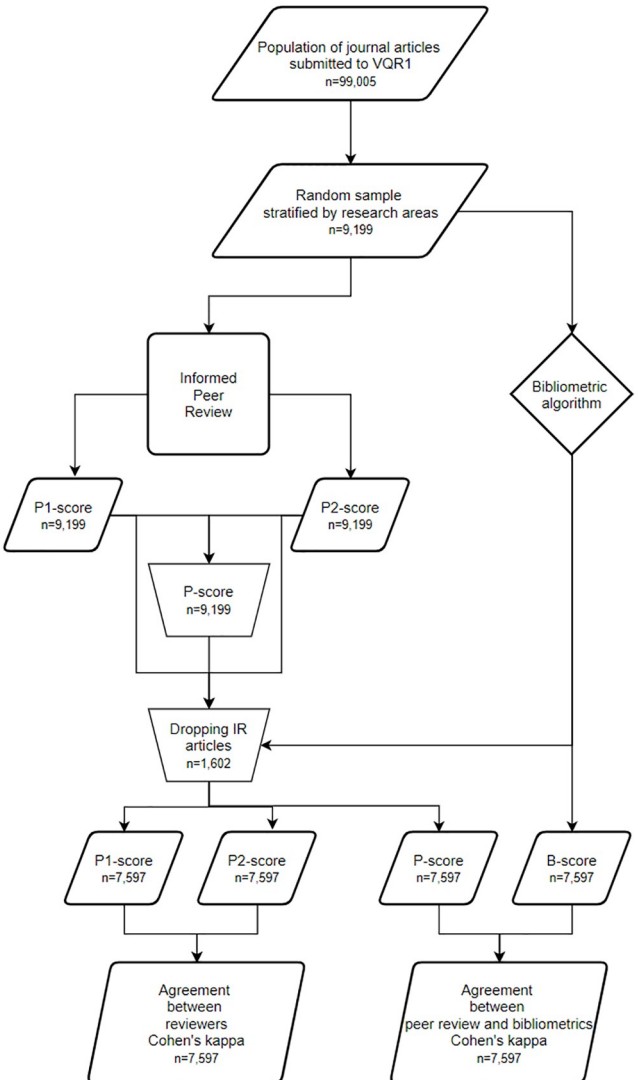

**Fig 1. Flowchart of EXP1.** Flowchart has been drawn with `diagrams.net` and by adopting its symbols and conventions.

the referee reports were received by the two members of the area panel, who formed a so-called "Consensus Group" (CG) for deciding the final score of the article (P-score).

In order to validate the dual system of evaluation, EXP1 and EXP2 considered only the "bibliometric areas" plus Area 13. They had a similar structure. Figs 1 and 2 report the flowcharts of the two experiments. The rationale of both experiments was very simple: a sample of the journal articles submitted to the research assessment was scored by the two methods of evaluations, i.e. through the bibliometric algorithm and IPR. In such a case, IPR was implemented by involving two reviewers, according to the same rules adopted in the research assessment. These raw data were then used for analyzing (i) the agreement between the evaluation obtained through IPR (P-score) and bibliometric algorithms (B-score) and (ii) the agreement between the scores decided by the two reviewers (P1-score and P2-score). The agreement between the scores is computed by using the weighted Cohen's kappa coefficient [22], a popular index of inter-rater agreement for nominal categories (see e.g. [23]). A high level of

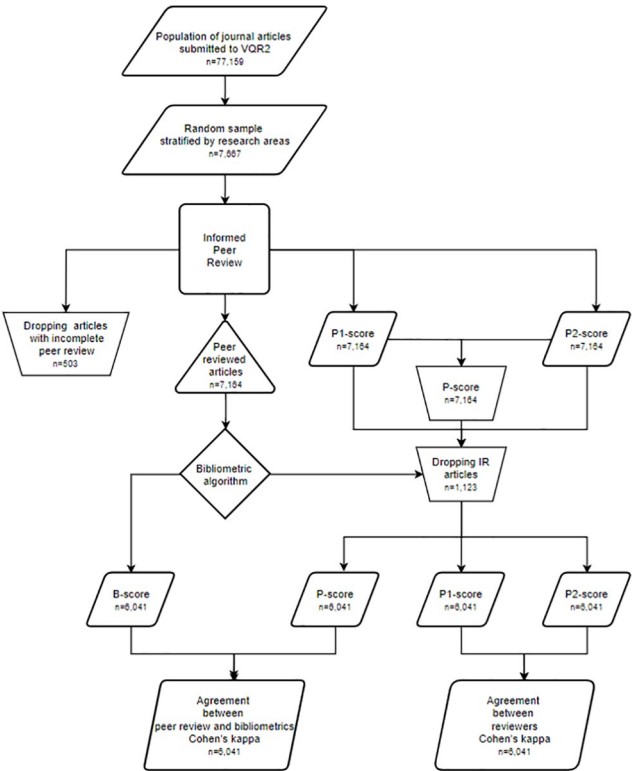

**Fig 2. Flowchart of EXP2.** Flowchart has been drawn with `diagrams.net` and by adopting its symbols and conventions.

agreement between IPR and bibliometric scores was interpreted as validating the dual method of evaluation.

EXP1 and EXP2 differed for a different timing of realization. EXP1 was done simultaneously with VQR1. Hence, the reviewers were unaware of partecipating to EXP1. Indeed, they were unable to distinguish between papers of the EXP1 sample and those they had to evaluate for the research assessment. The unique exception was Area 13, where panelists and referees knew that all the journal articles belonged to the EXP1 sample—since all the journal articles for the research assessment were evaluated automatically according to the journal ranking [6]. In contrast, EXP2 started after the conclusion of the activities of the research assessment. Therefore, panelists and reviewers knew that they were partecipating to EXP2. A second consequence of the different timing was that in EXP1 all the papers of the sample were peer-reviewed, since the successful administrative completion of the research assessment required the evaluation of all submitted articles. On the contrary, in EXP2 some papers did not receive a peer-review evaluation since some reviewers refused to do it. Therefore, in EXP2 there were missing data in the sample, which were not accounted for by ANVUR when the concordance indexes were computed.

## 4 Measures of agreement, sampling and data

The first step of this works consists in replicating ANVUR's computations. This end entails the adoption of the measure of agreement chosen by ANVUR. ANVUR used the Cohen's kappa coefficient and its weighted generalization, a commonly-adopted measure of agreement between classifications of two raters [22, 24]. Despite the Cohen's kappa coefficient is criticized

**Table 1. Population, sample and sub-sample sizes for scientific areas in EXP1.**

| Scientific Areas | Population | Sample | Sub-sample |
|---|---|---|---|
| Area 1—Mathematics and Informatics | 6758 | 631 | 438 |
| Area 2—Physics | 15029 | 1412 | 1212 |
| Area 3—Chemistry | 10127 | 927 | 778 |
| Area 4—Earth Sciences | 5083 | 458 | 377 |
| Area 5—Biology | 14043 | 1310 | 1058 |
| Area 6—Medicine | 21191 | 1984 | 1602 |
| Area 7—Agricultural and Veterinary Sciences | 6284 | 532 | 425 |
| Area 8a—Civil Engineering | 2460 | 225 | 198 |
| Area 9—Industrial and Information Engineering | 12349 | 1130 | 919 |
| Area 13—Economics and Statistics | 5681 | 590 | 590 |
|  | 99005 | 9199 | 7597 |

Source: ANVUR (2013, Appendix B).

for some methodological drawbacks (for more details, see [25, 26] among others), practitioners often adopt this index in order to assess the inter-rater agreement for categorical ratings, while its weighted counterpart is preferred when the categories can be considered ordinal (see e.g. [27, p. 548] and [28, p. 596]). Rough guidelines for interpreting Cohen's kappa values are available and a survey is provided by [6]. The guideline generally adopted is the one by Fagerland et al. [27, p. 550]—based on Landis and Koch [29], and slightly modified by Altman [30].

The replication of ANVUR's computations is surely useful, albeit not sufficient to reach a complete appreciation of the results of the two experiments. Indeed, some research questions should be carefully addressed. For EXP1 and EXP2, ANVUR [12, 17, Appendix B] adopted a stratified random sampling, where the target population was constituted by the journal articles submitted to the two research assessments. The sizes of the article populations in EXP1 and EXP2 are 99,005 and 77,159, respectively. The sample size was about 10% of the population size, i.e. 9,199 and 7,667 articles for EXP1 and EXP2, respectively. The stratified random samples were proportionally allocated with respect to the sizes of the research areas. The sizes of the strata in EXP1 and EXP2 are reported in Tables 1 and 2. Indeed, the Final Report remarks

**Table 2. Population, sample, sub-sample sizes and number of missing articles for scientific areas in EXP2.**

| Scientific Areas | Population | Sample | Sub-sample | Missing |
|---|---|---|---|---|
| Area 1—Mathematics and Informatics | 4631 | 467 | 344 | 23 |
| Area 2—Physics | 10182 | 1018 | 926 | 10 |
| Area 3—Chemistry | 6625 | 662 | 549 | 9 |
| Area 4—Earth Sciences | 3953 | 394 | 320 | 6 |
| Area 5—Biology | 10423 | 1037 | 792 | 86 |
| Area 6—Medicine | 15400 | 1524 | 1071 | 231 |
| Area 7—Agricultural and Veterinary Sciences | 6354 | 638 | 489 | 8 |
| Area 8b—Civil Engineering | 2370 | 237 | 180 | 3 |
| Area 9—Industrial and Information Engineering | 9930 | 890 | 739 | 108 |
| Area 11b—Psychology | 1801 | 180 | 133 | 5 |
| Area 13—Economics and Statistics | 5490 | 512 | 498 | 14 |
|  | 77159 | 7667 | 6041 | 503 |

Source: ANVUR (2017, Appendix B).

that: "The sample was stratified according to the distribution of the products among the sub-areas of the various areas" [17, Appendix B, p. 1, our translation]. For EXP1 results were published at a sub-area level, while for EXP2 results were solely published for areas. Moreover, the raw data at the sub-area level are not yet available.

A first research question deals with the statistical methodology adopted in the experiments. From this perspective, the two experiments were actually implemented in a design-based framework. Hence, their analysis requires a correct inferential setting in order to obtain the estimates of the considered concordance measures. To this aim, in Section 4 the model-based estimation of the weighted Cohen's kappa coefficient is revised and the design-based estimation of this coefficient is originally developed. On the basis of these theoretical results, it is possible to check if ANVUR's estimates of agreement are correct. In particular, ANVUR's estimates of Cohen's kappa coefficients and the corresponding confidence intervals may be compared with the appropriate design-based counterparts.

ANVUR computed the final results of EXP1 and EXP2 by solely considering a sub-sample of articles—and not the whole sample. This is illustrated in Figs 1 and 2 where the sizes of the populations, of the samples and of the final subsamples are reported. Indeed, ANVUR dropped from the computation of the concordance indexes the articles with an inconclusive bibliometric score IR, i.e. those receiving an IPR evaluation albeit they were not considered for agreement estimation. For EXP1, the reduction of the sample due to the exclusion of the paper classified as IR was not disclosed neither in ANVUR's official reports nor in [4]. Tables 1 and 2 reports the sizes of the sub-samples for EXP1 and EXP2, respectively. The exclusion of the IR papers might have boosted the value of the agreement measures, as argued by Baccini and De Nicolao [8]. The conjecture sounds as follows. ANVUR removed from EXP1 the most problematic articles for which the bibliometric algorithm was unable to reach a score. It cannot be excluded that these articles were also particularly difficult to evaluate for peer reviewers. Hence, ANVUR calculated the agreement indicators on sub-samples of articles that were "more favorable" to agreement than the complete samples.

The second research question, therefore, deals with the adoption of concordance measures which take into account the number of the IR articles which ANVUR dropped, as well as the number of missing articles. Actually, these articles could be ideally considered as belonging to a rating category for which agreement is not required. In such a case, there exist alternative variants of the weighted Cohen's kappa, which may suitably manage this issue. Hence, in Section 5 the design-based estimation of these variants of the weighted Cohen's kappa are also developed. In turn, in Section 6 the resulting point estimates and the corresponding confidence intervals are computed for EXP1 and EXP2, respectively.

A third and last question—which is limited to EXP2—deals with the distribution of missing papers per research area, i.e. those papers for which a peer review score is unavailable. As previously remarked, Table 2 reports the number of missing papers per area. Actually, drawbacks would arise if the distribution of missing articles in the sample occurred in a non-proportional way between the strata, since in this case some research areas could be more represented than others. ANVUR [17] claimed that this was not the case. Thus, in Section 7 a new testing procedure for the homogeneity of missing proportions between strata is developed and applied to EXP2 data.

These three questions are addressed by using the raw data of the two ANVUR experiments. The articles in each database have a unique anonymous identifier. For each article, the available variables are: (i) the research area; (ii) the bibliometric score (B); (iii) the score assigned by the first reviewer (P1); (iv) the score assigned by the second reviewer (P2); (v) the syntetic peer-review score (P). Data are available as S1 File (downloadable also from https://doi.org/10.5281/zenodo.3727460).

## 5 Design-based estimation of the Cohen's kappa coefficient

As anticipated, both EXP1 and EXP2 adopted the weighted Cohen's kappa coefficient as measure of agreement. In order to introduce our proposal for the design-based estimation of the Cohen's kappa coefficient, first it is instructive to revise its model-based counterpart (for a general discussion on the two paradigms, see e.g. [31]). In the model-based approach, two potential raters classify items into $c$ categories, which are labeled on the set $I = \{1, \ldots, c\}$ without loss of generality. The couple of evaluations given by the raters for an item is modeled as a bivariate random vector, say $(U, V)$, which takes values on the set $I \times I$. More appropriately, $(U, V)$ should defined as a random element, since categories are indexed by the first $c$ integers for the sake of simplicity—even if they are just labels. The joint probability function of $(U, V)$ is assumed to be

$$P(U = l, V = m) = \vartheta_{l,m} \ ,$$

where $l, m \in I$, while $\vartheta_{l,m} \geq 0$ and $\sum_{l=1}^{c} \sum_{m=1}^{c} \vartheta_{l,m} = 1$. Hence, the parameter space for the underlying model is actually given by

$$\mathcal{P} = \{\vartheta_{l,m} : \vartheta_{l,m} \geq 0, l, m \in I, \sum_{m=1}^{c} \sum_{m=1}^{c} \vartheta_{l,m} = 1\} \ .$$

Moreover, it holds that

$$P(U = l) := \vartheta_{l+} = \sum_{m=1}^{c} \vartheta_{l,m} \ , \ P(V = m) := \vartheta_{+m} = \sum_{l=1}^{c} \vartheta_{l,m}$$

are the marginal probability distributions of $U$ and $V$, respectively. In practice, $\vartheta_{l,m}$ represents the probability that an item be classified into the $l$-th category according to the first rating and into the $m$-th category according to the second rating. Similarly, $\vartheta_{l+}$ and $\vartheta_{+l}$ are the probabilities that the item be categorized into the $l$-th category according to the first rating and the second rating, respectively. Hence, the definition of the weighted Cohen's kappa in the model-based approach is given by

$$\kappa_{w,M} = \frac{\vartheta_o - \vartheta_e}{1 - \vartheta_e} \ ,$$

where

$$\vartheta_o = \sum_{l=1}^{c} \sum_{m=1}^{c} w_{lm} \vartheta_{lm} \ , \ \vartheta_e = \sum_{l=1}^{c} \sum_{m=1}^{c} w_{lm} \vartheta_{l+} \vartheta_{+m} \ ,$$

while the $w_{lm}$'s are weights which are suitably chosen in order to consider the magnitude of disagreement (see e.g. [27, p. 551]). In particular, the (usual) unweighted Cohen's kappa coefficient is obtained when $w_{lm} = 1$ if $l = m$ and $w_{lm} = 0$, otherwise.

In order to estimate the weighted Cohen's kappa under the model-based approach, let us assume that a random sample, say $(U_1, V_1), \ldots, (U_n, V_n)$, of $n$ copies of $(U, V)$ is available. Thus, the maximum-likelihood estimators of the $\vartheta_{lm}$'s, the $\vartheta_{l+}$'s and the $\vartheta_{+l}$'s are readily seen as

$$\hat{\vartheta}_{lm} = \frac{1}{n} \sum_{i=1}^{n} \mathbf{1}_{\{l\}}(U_i)\mathbf{1}_{\{m\}}(V_i) \ , \ \hat{\vartheta}_{l+} = \frac{1}{n} \sum_{i=1}^{n} \mathbf{1}_{\{l\}}(U_i) \ , \ \hat{\vartheta}_{+l} = \frac{1}{n} \sum_{i=1}^{n} \mathbf{1}_{\{l\}}(V_i) \ ,$$

where $\mathbf{1}_B$ is the usual indicator function of a set $B$, i.e. $\mathbf{1}_B(u) = 1$ if $u \in B$ and $1_B(u) = 0$, otherwise. Thus, on the basis of the invariance property of the maximum-likelihood estimation (see

e.g. Theorem 7.2.10 by Casella and Berger [32]), the maximum-likelihood estimator of $\kappa_{w,M}$ is provided by

$$\hat{\kappa}_{w,M} = \frac{\hat{\vartheta}_o - \hat{\vartheta}_e}{1 - \hat{\vartheta}_e} \ ,$$

where

$$\hat{\vartheta}_o = \sum_{l=1}^{c}\sum_{m=1}^{c} w_{lm}\hat{\vartheta}_{lm} \ , \ \hat{\vartheta}_e = \sum_{l=1}^{c}\sum_{m=1}^{c} w_{lm}\hat{\vartheta}_{l+}\hat{\vartheta}_{+m} \ .$$

Actually, $\hat{\kappa}_{w,M}$ is the weighted Cohen's kappa estimator commonly adopted in practical applications. Finally, it should remarked that the variance of $\hat{\kappa}_{w,M}$ is usually estimated by means of large-sample approximations [41, p. 610].

Under the design-based approach, there exists a fixed population of $N$ items which are classified into the $c$ categories on the basis of two ratings. Hence, the $j$-th item of the population is categorized according to the first evaluation—say $u_j \in I$—and the second evaluation—say $v_j \in I$—for $j = 1, \ldots, N$. It should be remarked that in this case the $N$ couples $(u_1, v_1), \ldots, (u_N, v_N)$ are fixed and given. Thus, the "population" weighted Cohen's kappa coefficient may be defined as

$$\kappa_w = \frac{p_o - p_e}{1 - p_e} \ , \tag{1}$$

where

$$p_o = \sum_{l=1}^{c}\sum_{m=1}^{c} w_{lm}p_{lm} \ , \ p_e = \sum_{l=1}^{c}\sum_{m=1}^{c} w_{lm}p_{l+}p_{+m} \ ,$$

while

$$p_{lm} = \frac{1}{N}\sum_{j=1}^{N}\mathbf{1}_{\{l\}}(u_j)\mathbf{1}_{\{m\}}(v_j) \ , \ p_{l+} = \frac{1}{N}\sum_{j=1}^{N}\mathbf{1}_{\{l\}}(u_j) \ , \ p_{+l} = \frac{1}{N}\sum_{j=1}^{N}\mathbf{1}_{\{l\}}(v_j) \ .$$

In this case, $p_{lm}$ is the proportion of items in the population classified into the $l$-th category according to the first rating and into the $m$-th category according to the second rating. Similarly, $p_{l+}$ and $p_{+l}$ are the proportions of items categorized into the $l$-th category according to the first rating and the second rating, respectively. Thus, for estimation purposes, the Cohen's kappa coefficient (1) is conveniently expressed as a smooth function of population totals—i.e. the $p_{lm}$'s, the $p_{l+}$'s and the $p_{+l}$'s. It is worth remarking that (1) is a fixed population quantity under the design-based approach, while its counterpart $\kappa_{w,M}$ under the model-based approach is a unknown quantity depending on the model parameters.

Let us now assume that a sampling design is adopted in order to estimate (1) and let us consider a sample of fixed size $n$. Moreover, let $S$ denote the set of indexes corresponding to the sampled items—i.e. a subset of size $n$ of the first $N$ integers—and let $\pi_j$ be the inclusion probability of the first order for the $j$-th item. As an example aimed to the subsequent application, let us assume that the population be partitioned into $L$ strata and that $N_h$ be the size of the $h$-th stratum with $h = 1, \ldots, L$. Obviously, it holds $N = \sum_{h=1}^{L} N_h$. If a stratified sampling design is considered, the sample is obtained by drawing $n_h$ items in the $l$-th stratum by means of simple random sampling without replacement in such a way that $n = \sum_{h=1}^{L} n_h$. Therefore, as is well known, it turns out that $\pi_j = n_h/N_h$ if the $j$-th item is in the $h$-th stratum (see e.g. [33]). When a

proportional allocation is adopted, it also holds that $n_h = nN_h/N$—and hence it obviously follows $\pi_j = n/N$.

In order to obtain the estimation of (1), it should be noticed that

$$\hat{p}_{lm} = \frac{1}{N} \sum_{j \in S} \frac{\mathbf{1}_{\{l\}}(u_j)\mathbf{1}_{\{m\}}(v_j)}{\pi_j} \ , \ \hat{p}_{l+} = \frac{1}{N} \sum_{j \in S} \frac{\mathbf{1}_{\{l\}}(u_j)}{\pi_j} \ , \ \hat{p}_{+l} = \frac{1}{N} \sum_{j \in S} \frac{\mathbf{1}_{\{l\}}(v_j)}{\pi_j} \ ,$$

are unbiased Horvitz-Thompson estimators of the population proportions $p_{lm}$, $p_{l+}$ and $p_{+l}$, respectively. Thus, by bearing in mind the general comments provided by Demnati and Rao [34] on the estimation of a function of population totals, a "plug-in" estimator of (1) is given by

$$\hat{\kappa}_w = \frac{\hat{p}_o - \hat{p}_e}{1 - \hat{p}_e} \ , \tag{2}$$

where

$$\hat{p}_o = \sum_{l=1}^{c}\sum_{m=1}^{c} w_{lm}\hat{p}_{lm} \ , \ \hat{p}_e = \sum_{l=1}^{c}\sum_{m=1}^{c} w_{lm}\hat{p}_{l+}\hat{p}_{+m} \ .$$

Even if estimator (2) is biased, its bias is negligible since (1) is a differentiable function of the population totals with non-null derivatives (for more details on such a result, see e.g. [33, p. 106]).

As usual, variance estimation is mandatory is order to achieve an evaluation of the accuracy of the estimator. Since (2) is a rather involved function of sample totals, its variance may be conveniently estimated by the linearization method or by the jackknife technique (see e.g. [34] and references therein). Alternatively, a bootstrap approach—which is based on a pseudo-population method—may be suitably considered (for more details on this topic, see e.g. [35].

It should be remarked that inconclusive ratings occur in EXP1 and EXP2 and—in addition—missing ratings are also present in EXP2. However, even if ANVUR does not explicitly states this issue, its target seems to be the sub-population of items with two reported ratings. Hence, some suitable variants of the Cohen's kappa coefficient have to be considered. In order to deal with an appropriate definition of the population parameter in this setting, the three suggestions provided by De Raadt et al. [36] could be adopted. For the sake of simplicity, let us suppose that inconclusive or missing ratings are classified into the $c$-th category. A first way to manage the issue consists in deleting all items which are not classified by both raters and apply the weighted Cohen's kappa coefficient to the items with two ratings (see also [37]). After some straightforward algebra, this variant of the population weighted Cohen's kappa coefficient may be written as

$$\kappa_w^{(1)} = \frac{p_o^{(1)} - p_e^{(1)}}{1 - p_e^{(1)}} \ , \tag{3}$$

where

$$p_o^{(1)} = \frac{\sum_{l=1}^{c-1}\sum_{m=1}^{c-1} w_{lm} p_{lm}}{\sum_{l=1}^{c-1}\sum_{m=1}^{c-1} p_{lm}} \ , \ p_e^{(1)} = \frac{\sum_{l=1}^{c-1}\sum_{m=1}^{c-1} w_{lm}(p_{l+} - p_{lc})(p_{+m} - p_{cm})}{\left(\sum_{l=1}^{c-1}\sum_{m=1}^{c-1} p_{lm}\right)^2} \ .$$

It is worth noting that (3) could be not a satisfactory index, since it does not take into account the size of inconclusive or missing ratings. Similarly to (1), its variant (3) can be

estimated as

$$\hat{\kappa}_w^{(1)} = \frac{\hat{p}_o^{(1)} - \hat{p}_e^{(1)}}{1 - \hat{p}_e^{(1)}} \quad, \tag{4}$$

where

$$\hat{p}_o^{(1)} = \frac{\sum_{l=1}^{c-1} \sum_{m=1}^{c-1} w_{lm} \hat{p}_{lm}}{\sum_{l=1}^{c-1} \sum_{m=1}^{c-1} \hat{p}_{lm}} \quad, \quad \hat{p}_e^{(1)} = \frac{\sum_{l=1}^{c-1} \sum_{m=1}^{c-1} w_{lm} (\hat{p}_{l+} - \hat{p}_{lc})(\hat{p}_{+m} - \hat{p}_{cm})}{\left( \sum_{l=1}^{c-1} \sum_{m=1}^{c-1} \hat{p}_{lm} \right)^2} \quad.$$

The second proposal by De Raadt et al. [36] for a variant of the weighted Cohen's kappa coefficient is based on Gwet's kappa [38]. The population weighted Gwet's kappa may be defined as

$$\kappa_w^{(2)} = \frac{p_o^{(2)} - p_e^{(2)}}{1 - p_e^{(2)}} \quad, \tag{5}$$

where

$$p_o^{(2)} = \frac{\sum_{l=1}^{c-1} \sum_{m=1}^{c-1} w_{lm} p_{lm}}{\sum_{l=1}^{c-1} \sum_{m=1}^{c-1} p_{lm}} \quad, \quad p_e^{(2)} = \frac{\sum_{l=1}^{c-1} \sum_{m=1}^{c-1} w_{lm} p_{l+} p_{+m}}{(1 - p_{c+})(1 - p_{+c})} \quad.$$

This index considers the sizes of inconclusive or missing ratings. Indeed, even if $p_o^{(2)} = p_o^{(1)}$, the quantity $p_e^{(2)}$ is actually a weighted sum of the products of type $p_{l+} p_{+l}$, in contrast to the quantity $p_e^{(1)}$ which is a weighted sum of the products of type $(p_{l+} - p_{lc})(p_{+m} - p_{cm})$. In turn, (5) may be estimated by means of

$$\hat{\kappa}_w^{(2)} = \frac{\hat{p}_o^{(2)} - \hat{p}_e^{(2)}}{1 - \hat{p}_e^{(2)}}, \tag{6}$$

where

$$\hat{p}_o^{(2)} = \frac{\sum_{l=1}^{c-1} \sum_{m=1}^{c-1} w_{lm} \hat{p}_{lm}}{\sum_{l=1}^{c-1} \sum_{m=1}^{c-1} \hat{p}_{lm}} \quad, \quad \hat{p}_e^{(2)} = \frac{\sum_{l=1}^{c-1} \sum_{m=1}^{c-1} w_{lm} \hat{p}_{l+} \hat{p}_{+m}}{(1 - \hat{p}_{c+})(1 - \hat{p}_{+c})} \quad.$$

The third proposal by De Raadt et al. [36] for a variant of (1) stems on assuming null weights for the inconclusive or missing ratings, i.e. by assuming that $w_{lm} = 0$ if $l = c$ or $m = c$. Hence, this variant is obviously defined as

$$\kappa_w^{(3)} = \frac{p_o^{(3)} - p_e^{(3)}}{1 - p_e^{(3)}}, \tag{7}$$

where

$$p_o^{(3)} = \sum_{l=1}^{c-1} \sum_{m=1}^{c-1} w_{lm} p_{lm} \quad, \quad p_e^{(3)} = \sum_{l=1}^{c-1} \sum_{m=1}^{c-1} w_{lm} p_{l+} p_{+m}.$$

In turn, (7) may be estimated by means of

$$\hat{\kappa}_w^{(3)} = \frac{\hat{p}_o^{(3)} - \hat{p}_e^{(3)}}{1 - \hat{p}_e^{(3)}}, \tag{8}$$

where

$$\hat{p}_o^{(3)} = \sum_{l=1}^{c-1}\sum_{m=1}^{c-1} w_{lm}\hat{p}_{lm} \ , \ \hat{p}_e^{(3)} = \sum_{l=1}^{c-1}\sum_{m=1}^{c-1} w_{lm}\hat{p}_{l+}\hat{p}_{+m} \ .$$

The previous findings are applied to the data collected in EXP1 and EXP2 in the following section.

## 6 Cohen's kappa coefficient estimation in the Italian experiments

The theoretical results presented in Section 5 can be applied to the raw data of the two experiments. Therefore, it is possible to implement appropriate estimates of the considered weighted Cohen's kappa coefficients for the agreement (i) between bibliometric (B) and peer-review ratings (P) and (ii) between the ratings of the first referee (P1) and the second referee (P2). The dot-plot graphics of the distributions of the ratings are provided as S1–S4 Figs.

Some preliminary considerations are required on the choice of the weights for the computation of Cohen's kappa. Let $\mathbf{W} = (w_{lm})$ generally denote the square matrix of order $c$ of the weights. The selection of the weights is completely subjective and the adoption of different sets of weights may obviously modify the concordance level. ANVUR presented results for two sets of weights in EXP1 and EXP2. The first set of weights consisted in the usual linear weights, i.e. $w_{lm} = 1 - |l - m|/(c - 1)$. In such a case, the matrices of linear weights for EXP1 and EXP2 are given, respectively, by

$$\mathbf{W} = \begin{pmatrix} 1 & \frac{2}{3} & \frac{1}{3} & 0 \\ \frac{2}{3} & 1 & \frac{2}{3} & \frac{2}{3} \\ \frac{2}{3} & \frac{2}{3} & 1 & \frac{2}{3} \\ 0 & \frac{2}{3} & \frac{2}{3} & 1 \end{pmatrix}$$

and

$$\mathbf{W} = \begin{pmatrix} 1 & \frac{3}{4} & \frac{1}{2} & \frac{1}{4} & 0 \\ \frac{3}{4} & 1 & \frac{3}{4} & \frac{1}{2} & \frac{1}{4} \\ \frac{1}{2} & \frac{3}{4} & 1 & \frac{3}{4} & \frac{1}{2} \\ \frac{1}{4} & \frac{1}{2} & \frac{3}{4} & 1 & \frac{3}{4} \\ 0 & \frac{1}{4} & \frac{1}{2} & \frac{3}{4} & 1 \end{pmatrix} .$$

The second set was originally developed by ANVUR and named "VQR-weights". The matrices of VQR-weights for EXP1 and EXP2 are respectively given by

$$\mathbf{W} = \begin{pmatrix} 1 & 0.8 & 0.5 & 0 \\ 0.8 & 1 & 0.7 & 0.2 \\ 0.5 & 0.7 & 1 & 0.5 \\ 0 & 0.2 & 0.5 & 1 \end{pmatrix}$$

and

$$\mathbf{W} = \begin{pmatrix} 1 & 0.7 & 0.4 & 0.1 & 0 \\ 0.7 & 1 & 0.7 & 0.4 & 0.3 \\ 0.4 & 0.7 & 1 & 0.7 & 0.6 \\ 0.1 & 0.4 & 0.7 & 1 & 0.9 \\ 0 & 0.3 & 0.6 & 0.9 & 1 \end{pmatrix}.$$

The VQR-weights were based on the scores adopted in the research assessments even if they appear counter-intuitive, since they attribute different weights to a same category distance. For example, in EXP1 a distance of two categories is weighted with 0.5 if it occurs for the first and the third category, while it is solely weighted with 0.2 if it occurs for the second and fourth category. In order to reproduce ANVUR's results, the sets of linear weights and VQR-weights are solely considered. In addition, for improving readability, the analysis and the comments are limited to the computation based on VQR-weights, while the results for linear weights are available as S1 Table.

At first, the estimation of (3), (5) and (7) are considered for the agreement of the bibliometric and peer-review ratings by means of the estimators (4), (6) and (8). The estimation was carried out for each area and for the global population in both EXP1 and EXP2. Variance estimation was carried out by means of the Horvitz-Thompson based bootstrap—stemming on the use of a pseudo-population—which is described by Quatember [35, p. 16 and p. 80]. The whole computation was implemented by means of the algebraic software `Mathematica` [39]. The corresponding `Mathematica` notebooks are available on request. The point and interval estimates are given in Tables 3 and 4. The columns labeled "ANVUR" report the point and interval estimates provided by ANVUR [12, 17]. Moreover, in Figs 3 and 4 the estimates (4), (6) and (8) and the corresponding confidence intervals at the 95% confidence level are plotted in the "error-bar" style.

**Table 3. Cohen's kappa coefficient estimates (percent) for EXP1 (95% confidence level intervals in parenthesis), bibliometric vs peer review ratings.**

| Area | ANVUR[a] | $\hat{\kappa}_w^{(1)}$ | $\hat{\kappa}_w^{(2)}$ | $\hat{\kappa}_w^{(3)}$ |
|------|----------|------------------------|------------------------|------------------------|
| 1 | 31.73(23.00,40.00) | 31.73(25.21,38.26) | 33.40(26.80,40.00) | 15.07(11.76,18.38) |
| 2 | 25.15(21.00,29.00) | 25.15(21.10,29.19) | 29.15(25.29,33.01) | 18.91(16.24,21.58) |
| 3 | 22.96(17.00,29.00) | 22.96(18.05,27.86) | 23.98(19.09,28.88) | 14.52(11.32,17.71) |
| 4 | 29.85(21.00,39.00) | 29.85(23.32,36.37) | 30.24(23.69,36.79) | 20.32(15.66,24.99) |
| 5 | 34.53(29.00,40.00) | 34.53(30.51,38.54) | 36.62(32.72,40.51) | 23.85(21.13,26.58) |
| 6 | 33.51(29.00,38.00) | 33.51(30.30,36.72) | 34.62(31.47,37.77) | 22.73(20.51,24.95) |
| 7 | 34.37(27.00,42.00) | 34.37(27.99,40.75) | 36.62(30.59,42.65) | 22.60(18.43,26.77) |
| 8a | 22.61(11.00,34.00) | 22.61(12.70,32.52) | 22.99(13.06,32.92) | 16.35(8.90,23.80) |
| 9 | 17.10(13.00,21.00) | 17.10(13.17,21.03) | 21.95(17.78,26.11) | 12.56(10.12,15.01) |
| 13 | 61.04(53.00,69.00)[b] | 54.17(49.37,58.98) | 54.17(49.37,58.98) | 54.17(49.37,58.98) |
| All | 38.00(36.00,40.00)[c] | 34.15(32.64,35.66) | 35.76(34.28,37.24) | 23.28(22.23,24.33) |

[a] Source: [12, Appendix B]. Reproduced in [4].

[b] Estimated with the wrong system of weights as documented in [8]. Benedetto et al. [15] justified it as "factual error in editing of the table" and published a corrected estimate of 54.17.

[c] Ancaiani et al. [4] reported a different estimate of 34.41, confirmed also in [15].

**Table 4. Cohen's kappa coefficient estimates (percent) for EXP2 (95% confidence level intervals in parenthesis), bibliometric vs peer review ratings.**

| Area | ANVUR[a] | $\hat{\kappa}_w^{(1)}$ | $\hat{\kappa}_w^{(2)}$ | $\hat{\kappa}_w^{(3)}$ |
|------|----------|------------------------|------------------------|------------------------|
| 1 | 21.50(15.10,27.80) | 21.48(15.38,27.58) | 22.85(16.71,29.00) | 14.97(11.79,18.16) |
| 2 | 26.50(22.40,30.50) | 26.48(22.61,30.34) | 28.66(24.86,32.46) | 22.35(19.46,25.23) |
| 3 | 19.50(14.30,24.70) | 19.49(14.60,24.38) | 20.85(16.01,25.69) | 13.71(10.71,16.72) |
| 4 | 23.90(16.60,31.20) | 23.90(17.02,30.77) | 24.52(17.75,31.28) | 15.78(11.55,20.01) |
| 5 | 24.10(19.70,28.40) | 24.07(19.98,28.15) | 25.01(20.97,29.05) | 19.93(17.75,22.11) |
| 6 | 22.80(19.50,26.20) | 22.83(19.62,26.04) | 24.47(21.32,27.62) | 21.00(19.49,22.51) |
| 7 | 27.00(21.30,32.70) | 27.01(21.66,32.36) | 28.76(23.56,33.96) | 16.02(13.05,18.99) |
| 8b | 17.20(8.80,25.60) | 17.21(9.183,25.23) | 20.36(12.55,28.16) | 11.45(7.23,15.67) |
| 9 | 16.90(12.90,21.00) | 16.91(13.04,20.78) | 19.62(15.82,23.42) | 18.51(16.58,20.44) |
| 11b | 24.10(13.70,34.50) | 24.09(14.30,33.88) | 25.45(15.93,34.97) | 14.76(9.556,19.95) |
| 13 | 30.90(26.20,35.50) | 30.85(26.36,35.34) | 30.85(26.36,35.34) | 31.54(27.51,35.57) |
| All | 26.00(24.50,27.60) | 26.10(24.64,27.56) | 27.31(25.87,28.74) | 20.88(20.05,21.71) |

[a] Source: [17, Appendix B].

Actually, the point estimates given by ANVUR correspond to those computed by means of (4). Thus, even if this issue is not explicitly stated in its reports [12, 17], ANVUR focused on the sub-population of articles with two reported ratings and considered the estimation of (3). Hence, the Cohen's kappa coefficient assumed by ANVUR does not account for the size of inconclusive ratings in EXP1, and for the size of inconclusive or missing ratings in EXP2. Moreover, the confidence intervals provided by ANVUR—and reported in Tables 3 and 4— are the same computed by means of the packages **psych** (in the case of EXP1) and **vcd** (in the case of EXP2) of the software R [40]. Unfortunately, these confidence intervals rely on the model-based approximation for large samples described by Fleiss et al. [41, p. 610]. Thus, even if ANVUR has apparently adopted a design-based inference, the variance estimation is carried out in a model-based approach. The columns corresponding to $\hat{\kappa}_w^{(1)}$ of Tables 3 and 4 show the appropriate version of ANVUR estimates, i.e. the design-based point estimates and the

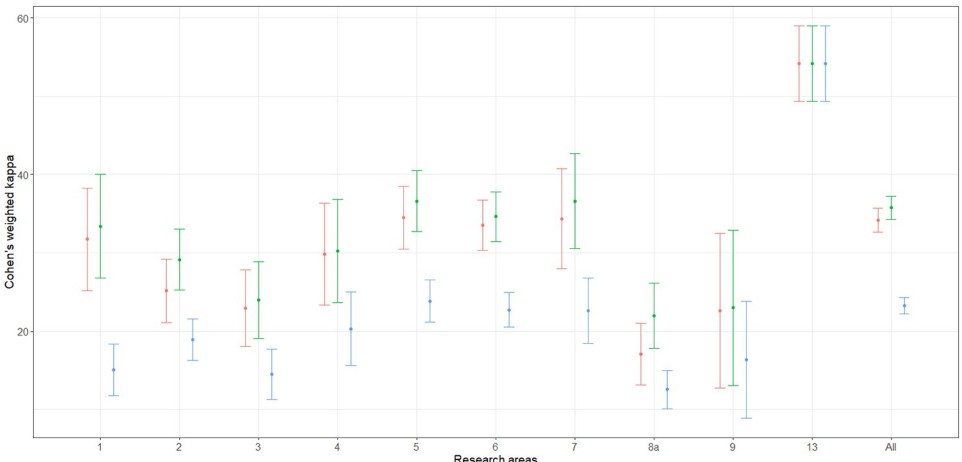

**Fig 3. "Error-bar" plots of the Cohen's kappa coefficient estimates (percent) for EXP1, bibliometric vs peer review ratings.** The confidence intervals are at 95% confidence level and estimates corresponding to $\hat{\kappa}_w^{(1)}$, $\hat{\kappa}_w^{(2)}$ and $\hat{\kappa}_w^{(3)}$ are in red, green and blue, respectively.

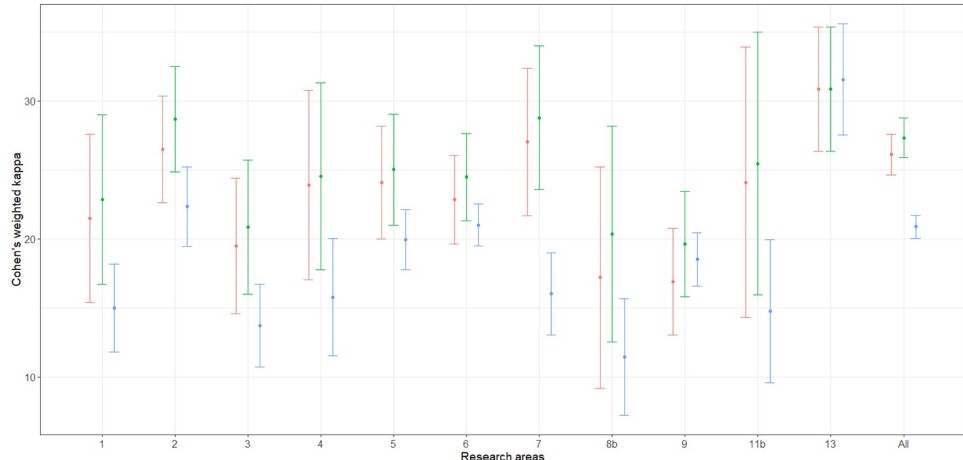

**Fig 4. "Error-bar" plots of the Cohen's kappa coefficient estimates (percent) for EXP2, bibliometric vs peer review ratings.** The confidence intervals are at 95% confidence level and estimates corresponding to $\hat{\kappa}_w^{(1)}$, $\hat{\kappa}_w^{(2)}$ and $\hat{\kappa}_w^{(3)}$ are in red, green and blue, respectively.

corresponding confidence intervals, which were computed by the bootstrap method. These confidence intervals are generally narrower than those originally computed by ANVUR—consistently with the fact that a stratified sampling design is carried out, rather than a simple random sampling design.

It is also convenient to consider the two alternative definitions of the weighted Cohen's kappa coefficient (5) and (7) and the corresponding estimators (6) and (8). These concordance measures take into account the sizes of the discarded articles—as formally explained in Section 5. From Tables 3 and 4, for both EXP1 and EXP2, the point and interval estimates corresponding to $\hat{\kappa}_w^{(2)}$ are similar to those corresponding to $\hat{\kappa}_w^{(1)}$. In contrast, the point and interval estimates corresponding to $\hat{\kappa}_w^{(3)}$ tend to be sistematically smaller than those corresponding to $\hat{\kappa}_w^{(1)}$. Arguably, this outcome should be expected. Indeed, (7) is likely to be more conservative than (3) and (5), since it assigns null weights to IR and missing articles.

By considering Fig 3, the first evidence is that Area 13—i.e. Economics and Statistics—is likely to be an outlier. In particular, point and interval estimates are identical when estimated by using (4), (6) or (8), since in Area 13 the use of simple journal ranking—as remarked in Section 3—did not produce IR score. More importantly, in Area 13 the value of agreement for EXP1 is higher than 0.50 and much higher than the values of all the other areas. Baccini and De Nicolao [6, 7] documented that in Area 13 the protocol of the experiment was substantially modified with respect to the other areas and contributed to boost the concordance between bibliometrics and peer review. In contrast, from Fig 4, Area 13 cannot be considered an outlier as in EXP1—even if it shows slightly higher values of agreement with respect to the other areas. Indeed, in EXP2 Area 13 adopted the same protocol of the other areas. Thus, it could be conjectured that the higher agreement was due to the exclusive use of journal ranking for attributing bibliometric scores.

Let us focus on the other areas in EXP1 and EXP2. The confidence intervals corresponding to $\hat{\kappa}_w^{(1)}$ and $\hat{\kappa}_w^{(2)}$ are largely overlapped. For most of the areas, the upper bound of the confidence intervals corresponding to $\hat{\kappa}_w^{(3)}$ is smaller than the lower bound of the confidence intervals corresponding to $\hat{\kappa}_w^{(1)}$ and $\hat{\kappa}_w^{(2)}$. Therefore, ANVUR's choice of discarding IR and missing articles presumably boosted the agreement between bibliometrics and peer review. Anyway, the upper bounds of the confidence intervals corresponding to $\hat{\kappa}_w^{(2)}$ are generally smaller than 0.40, and

**Table 5. Cohen's kappa coefficient estimates (percent) for EXP1 (95% confidence level intervals in parenthesis), P1 vs P2 ratings.**

| Area | ANVUR[a] | $\hat{\kappa}_w^{(1)}$ | $\hat{\kappa}_w^{(1)}$ (DBR)[e] | $\hat{\kappa}_w^{(1)}$ (IR)[f] |
|------|----------|------------------------|----------------------------------|--------------------------------|
| 1 | 35.16(26.00,44.00) | 33.31(27.54,39.09) | 35.16(25.26,45.06) | 28.87(18.17,39.57) |
| 2 | 22.71(18.00,28.00) | 23.42(19.44,27.41) | 22.71(17.28,28.14) | 19.31(9.227,29.39) |
| 3 | 23.81(17.00,30.00) | 20.83(16.00,25.65) | 23.81(17.73,29.89) | 2.56(-7.01,12.15) |
| 4 | 25.48(15.00,36.00) | 23.27(16.55,30.00) | 25.48(16.59,34.36) | 12.37(-3.47,28.23) |
| 5 | 27.17(21.00,33.00) | 24.85(20.76,28.93) | 27.17(21.56,32.78) | 11.12(1.77,20.46) |
| 6 | 23.56(19.00,29.00) | 21.85(18.57,25.12) | 23.56(19.09,28.03) | 11.84(4.19,19.48) |
| 7 | 26.56(21.00,33.00)[b] | 17.47(11.34,23.61) | 16.99(8.15,25.83) | 16.41(2.91,29.90) |
| 8a | 19.43(6.00,32.00) | 19.92(9.64,30.21) | 19.43(6.65,32.20) | 23.77(-7.45,54.99) |
| 9 | 18.18(12.00,24.00) | 19.39(14.93,23.84) | 18.18(11.72,24.64) | 21.1(10.70,31.50) |
| 13 | 45.99(38.00,54.00)[c] | 38.98(33.50,44.47) | 38.98(33.50,44.47) | - |
| All | 33.00(31.00,35.00)[d] | 26.68(25.16,28.20) | 27.92(25.90,29.95) | 18.90(15.30,22.50) |

[a] Source: [12]. Reproduced in [4].

[b] Estimated with the wrong system of weights as reported in [8]. Benedetto et al. [15] justified it as "factual error in editing of the table" and published a corrected estimate of 16.99.

[c] Estimated with the wrong system of weights as reported in [8]. Benedetto et al. [15] justified it as "factual error in editing of the table" and published a corrected estimate of 38.998.

[d] Ancaiani et al. [4] reported a different estimate of 28.16, confirmed also in [15].

[e] Weighted Cohen's kappa for the sets of articles with a definite bibliometric rating (DBR).

[f] Weighted Cohen's kappa for the sets of articles without a definite bibliometric rating and submitted to informed peer review (IPR).

those corresponding to $\hat{\kappa}_w^{(3)}$ are generally smaller than 0.30. A baseline for interpreting these values is provided in Table.13.6 by Fagerland et al. [27, p. 550]. According to this guideline, a value of the simple Cohen's kappa less than or equal to 0.20 is considered as a "poor" concordance and a value in the interval (0.20, 0.40) is considered as a "weak" concordance; values in the intervals (0.40, 0.60] and (0.60, 1.00] are considered respectively as indicating a "moderate" and a "very good" concordance. However, it should be remarked that these considerations are carried out for the simple Cohen's kappa. Hence, the small values of the weighted Cohen's kappa coefficients can be interpreted as indicating a concordance even worse than weak.

Subsequently, it is also considered the estimation of the Cohen's kappa coefficient for the agreement of the ratings attributed to the articles by the two reviewers, i.e. P1 and P2.

Thus, the estimation of (3) is computed for the population of articles, for the sub-population of articles receiving a Definite Bibliometric Rating (DBR) and for the sub-population of articles with an Inconclusive bibliometric Rating (IR). The point and interval estimates are reported in Tables 5 and 6, and displayed in Figs 5 and 6 in "error-bar" style. It should be remarked that—owing to the use of journal ranking—there are no IR articles for Area 13.

In Tables 5 and 6, the column labeled "ANVUR" reports the estimates provided by ANVUR [12, 17]. In turn, ANVUR did not explicitly state that it aimed to estimate (3) in the sub-population of articles with a definite bibliometric rating. However, this issue can be inferred from Tables 5 and 6, where—unless specific errors in the ANVUR computation for some areas—the ANVUR point estimates correspond to $\hat{\kappa}_w^{(1)}$ for the sub-population DBR. The confidence intervals provided by ANVUR are the same computed by means of the packages **psych** and **vcd** of the software R [40]. Thus, in this case also, ANVUR has apparently adopted a design-based inference, even if variance estimation is carried out in a model-based approach. Therefore, in Tables 5 and 6 the column corresponding to $\hat{\kappa}_w^{(1)}$ for the sub-population DBR reports the appropriate version of ANVUR point and interval estimates. The point estimate of

**Table 6. Cohen's kappa coefficient estimates (percent) for EXP2 (95% confidence level intervals in parenthesis), P1 vs P2 rating.**

| Area | ANVUR[a] | $\hat{\kappa}_w^{(1)}$ | $\hat{\kappa}_w^{(1)}$ (DBR)[b] | $\hat{\kappa}_w^{(1)}$ (IR)[c] |
|------|----------|------------------------|--------------------------------|-------------------------------|
| 1 | 20.20(12.90,27.50) | 23.92(17.68,30.15) | 20.18(11.11,29.25) | 35.71(22.11,49.31) |
| 2 | 19.50(14.60,24.40) | 21.13(16.75,25.52) | 19.50(14.24,24.77) | 20.29(6.81,33.77) |
| 3 | 14.00(7.90,20.10) | 14.67(9.36,19.98) | 13.99(7.14,20.83) | 15.53(3.04,28.02) |
| 4 | 18.90(11.10,26.80) | 18.63(11.97,25.29) | 18.94(10.14,27.75) | 12.4(-2.21,27.02) |
| 5 | 19.50(14.60,24.50) | 20.21(15.80,24.63) | 19.53(13.65,25.41) | 20.73(9.82,31.63) |
| 6 | 19.10(17.90,23.20) | 17.84(14.24,21.44) | 19.08(14.29,23.87) | 7.69(-0.73,16.13) |
| 7 | 19.60(13.40,25.80) | 22.38(17.14,27.63) | 19.57(11.58,27.57) | 28.34(17.54,39.15) |
| 8b | 3.50(-0.06,13.20) | 8.70(0.22,17.19) | 3.47(-9.42,16.37) | 22.41(5.90,38.92) |
| 9 | 15.10(9.90,20.30) | 15.36(10.84,19.89) | 15.09(8.87,21.31) | 12.71(1.65,23.76) |
| 11b | 25.70(13.30,38.20) | 25.79(15.39,36.19) | 25.72(8.93,42.50) | 20.68(0.22,41.15) |
| 13 | 31.20(25.40,36.90) | 31.15(25.69,36.61) | 31.15(25.69,36.61) | - |
| All | 23.40(21.60,25.20) | 23.54(21.97,25.10) | 23.50(21.48,25.52) | 19.85(15.94,23.77) |

[a] Source: [17].

[b] Weighted Cohen's kappa for the sets of articles with a definite bibliometric rating (DBR).

[c] Weighted Cohen's kappa for the sets of articles without a definite bibliometric rating and submitted to informed peer review (IPR).

(3) between the two reviewers for the population of articles, i.e. the column corresponding to $\hat{\kappa}_w^{(1)}$ in Tables 5 and 6, is generally lower then 0.30 with the exception of Area 13. The confidence intervals corresponding to $\hat{\kappa}_w^{(1)}$ overlap with the confidence intervals corresponding to $\hat{\kappa}_w^{(1)}$ for the sub-population DBR. From Figs 5 and 6, it is also apparent that $\hat{\kappa}_w^{(1)}$ is generally greater than $\hat{\kappa}_w^{(1)}$ for the sub-population IR. This last issue confirms the conjecture by Baccini and De Nicolao [8] that articles for which bibliometric rating was inconclusive were also the more difficult to evaluate for reviewers, by showing a smaller degree of agreement for these papers.

For both experiments, ANVUR directly compared the concordances between P1 and P2 with the ones between peer review and bibliometrics (see [9, p. 8] for a critique to this

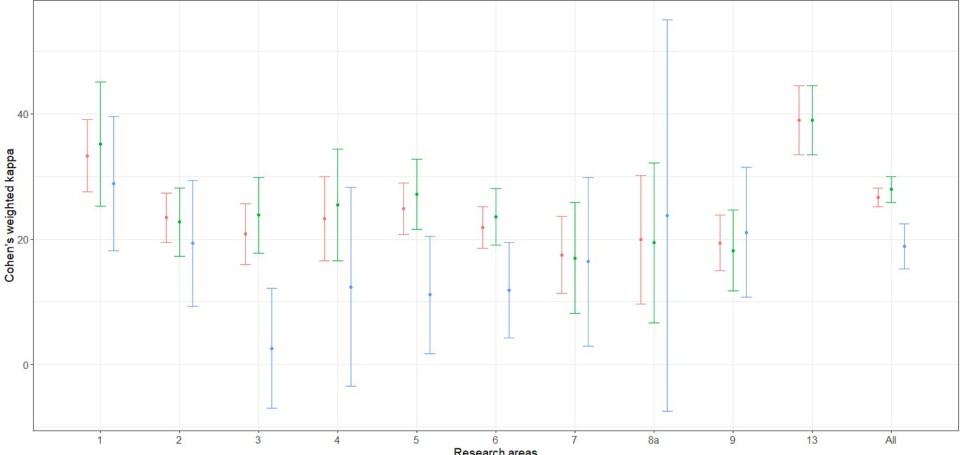

**Fig 5. "Error-bar" plots of the Cohen's kappa coefficient estimates (percent) for EXP1, P1 vs P2 ratings.** The confidence intervals are at 95% confidence level and estimates corresponding to $\hat{\kappa}_w^{(1)}$, $\hat{\kappa}_w^{(1)}$ (DBR) and $\hat{\kappa}_w^{(1)}$ (IR) are in red, green and blue, respectively.

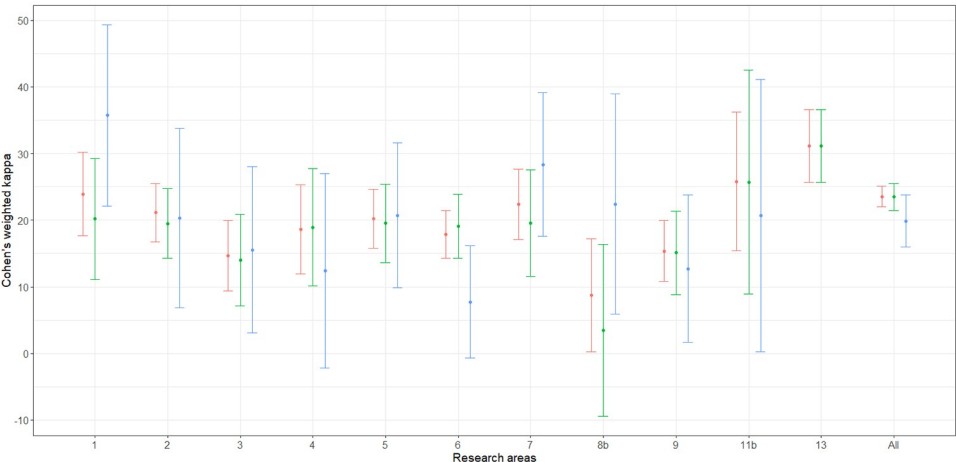

**Fig 6. "Error-bar" plots of the Cohen's kappa coefficient estimates (percent) for EXP2, P1 vs P2 ratings.** The confidence intervals are at 95% confidence level and estimates corresponding to $\hat{\kappa}_w^{(1)}$, $\hat{\kappa}_w^{(1)}$ (DBR) and $\hat{\kappa}_w^{(1)}$ (IR) are in red, green and blue, respectively.

comparison). As for EXP1, Ancaiani et al. commented the results of the comparison as follows: "the degree of concordance among different reviewers is generally lower than that obtained between the aggregate peer review and the bibliometric evaluation: in this sense, combining evaluations obtained with peer review and bibliometric methods can be considered as more reliable than the usual practice of combining two or more different evaluations obtained by various reviewers of the same article" [4]. Actually, they compared the level of agreement between bibliometrics and peer review (i.e. column ANVUR in Table 3) with the agreement of the two referees for the sub-population DBR (more precisely, column ANVUR in Table 5). When the appropriate estimates are considered, i.e. the second column in Table 5, it is apparent that Ancaiani et al.'s statement is no longer true. Hence, their policy suggestion cannot be considered as evidence-based. Actually Ancaiani et al statement appears true only for Area 13, where the concordance indexes between bibliometrics and peer review are much higher than the corresponding indexes between the two reviewers (see Tables 3 and 5). Also in this case, the exception of Area 13 is probably due to the modification of the protocol of the experiment that boosted the agreement between peer review and bibliometrics.

As to EXP2, the agreement between the two reviewers is similar to the agreement between bibliometrics and peer review—even in Area 13 where the experiment was implemented with a protocol identical to the other areas. These estimates are at odds with ANVUR conclusions: "It is particularly important the result that the degree of agreement between the bibliometric and the peer evaluation is always higher than the one existing between the two individual peer reviews" [17]. Also in this case, ANVUR conclusions were based on estimates computed on the sub-population of articles that boosted—as previously remarked—the values of agreement between bibliometrics and peer review.

## 7 Testing homogeneity of missing proportions between strata

In the case of EXP2, Section 4 considers the sizes of missing peer ratings as fixed and—accordingly—a design-based approach for the estimation of rating agreement is carried out. However, it could be also interesting to assess the homogeneity of missing proportions in the different areas by assuming a random model for the missing peer ratings, i.e. by considering a model-based approach for missing proportion estimation and testing. In order to provide an

appropriate setting in such a case, let us suppose in turn a fixed population of $N$ items partitioned into $L$ strata. Moreover, a stratified sampling design is adopted and the notations introduced in Section 2 are assumed. Hence, each item in the $h$-th stratum may be missed with probability $\theta_h \in [0, 1]$—independently with respect to the other items. Thus, the size of missing items in the $h$-th stratum, say $M_h$, is a random variable (r.v.) distributed according to the Binomial law with parameters $N_h$ and $\theta_h$, i.e. the probability function (p.f.) of $M_h$ turns out to be

$$p_{M_h}(m) = \binom{N_h}{m} \theta_h^m (1 - \theta_h)^{N_h - m} \mathbf{1}_{\{0,1,\ldots,N_h\}}(m) \ .$$

Let us assume that the r.v. $X_h$ represents the size of missing items of the $h$-th stratum in the sample. By supposing that the items are missed independently with respect to the sampling design, the distribution of the r.v. $X_h$ given the event $\{M_h = m\}$ is the Hypergeometric law with parameters $n_h$, $m$ and $N_h$, i.e. the corresponding conditioned p.f. is given by

$$p_{X_h | \{M_h = m\}}(x) = \frac{\binom{m}{x}\binom{N_h - m}{n_h - x}}{\binom{N_h}{n_h}} \mathbf{1}_{\{\max(0, n_h - N_h + m), \ldots, \min(n_h, m)\}}(x).$$

Hence, on the basis of this finding and by using the result by Johnson et al. [42, p. 377], the r.v. $X_h$ is distributed according to the Binomial law with parameters $n_h$ and $\theta_h$, i.e. the p.f. of $X_h$ is

$$p_{X_h}(x) = \binom{n_h}{x} \theta_h^x (1 - \theta_h)^{n_h - x} \mathbf{1}_{\{0,1,\ldots,n_h\}}(x)$$

for each $h = 1, \ldots, L$. Obviously, the $X_h$'s are independent r.v.'s.

Under the frequentist paradigm, let us consider the null hypothesis of missing proportion homogeneity $H_0: \theta_h = \theta, \forall h = 1, \ldots, L$, versus the alternative hypothesis $H_1: \theta_h \neq \theta, \exists h = 1, \ldots, L$. For a given $(x_1, \ldots, x_L) \in \mathbb{N}^L$ such that $y = \sum_{h=1}^L x_h$, the likelihood function under the null hypothesis is given by

$$L_0(\theta) \propto \theta^y (1 - \theta)^{n - y} \mathbf{1}_{[0,1]}(\theta) \ ,$$

while the likelihood function under the alternative hypothesis is given by

$$L_1(\theta_1, \ldots, \theta_L) \propto \prod_{h=1}^L \theta_h^{x_h} (1 - \theta_h)^{n_h - x_h} \mathbf{1}_{[0,1]^L}(\theta_1, \ldots, \theta_L) \ .$$

Thus, the likelihood estimator of $\theta$ under the null hypothesis turns out to be $\hat{\theta} = Y/n$, where $Y = \sum_{h=1}^L X_h$. In addition, the likelihood estimator of $(\theta_1, \ldots, \theta_L)$ under the alternative hypothesis turns out to be $(\hat{\theta}_1, \ldots, \hat{\theta}_L)$, where $\hat{\theta}_h = X_h/n_h$.

The likelihood-ratio test statistic could be adopted in order to assess the null hypothesis. However, in the present setting the large-sample results are precluded, since the sample size $n$ is necessarily bounded by $N$ and the data sparsity could reduce the effectiveness of the large-sample approximations. A more productive approach may be based on conditional testing (see e.g. [43, Chapter 10]). First, it is considered the $\chi^2$ test statistic—asymptotically equivalent in distribution to the likelihood-ratio test statistic—which in this case, after some algebra, reduces

to

$$R := R(X_1, \ldots, X_L) = \sum_{h=1}^{L} \frac{n_h(\hat{\theta}_h - \hat{\theta})^2}{\hat{\theta}(1 - \hat{\theta})} \ .$$

It should be remarked that the r.v. $Y$ is sufficient for $\theta$ under the null hypothesis. Hence, the distribution of the random vector $(X_1, \ldots, X_L)$ given the event $\{Y = y\}$ does not depend on $\theta$. Moreover, under the null hypothesis, the distribution of the random vector $(X_1, \ldots, X_L)$ given the event $\{Y = y\}$ is the multivariate Hypergeometric law with parameters $y$ and $(n_1, \ldots, n_L)$, i.e. the corresponding conditioned p.f. is

$$p_{(X_1,\ldots,X_L)|\{Y=y\}}(x_1, \ldots, x_L) = \frac{\prod_{h=1}^{L} \binom{n_h}{x_h}}{\binom{n}{y}} \mathbf{1}_A(x_1, \ldots, x_L),$$

where

$$A = \left\{ (x_1, \ldots, x_L) \in \mathbb{N}^L : x_h \in \{ \max(0, n_h - n + y), \ldots, \min(n_h, y) \}, \sum_{h=1}^{L} x_h = y \right\} \ .$$

Thus, by assuming the conditional approach, an exact test may be carried out. Indeed, if $r$ represents the observed realization of the test statistic $R$, the corresponding $P$-value is

$$P(R \geq r \mid \{Y = y\}) = \sum_{(x_1,\ldots,x_L) \in C_r} p_{(X_1,\ldots,X_L)|\{Y=y\}}(x_1, \ldots, x_L) \ ,$$

where $C_r = \{(x_1, \ldots, x_L) \in A : R(x_1, \ldots, x_L) \geq r\}$. It should be remarked that the previous $P$-value may be approximated by means of a Monte Carlo method by generating realizations of a Hypergeometric random vector with parameters $y$ and $(n_1, \ldots, n_L)$. The generation of each realization requires $(L - 1)$ Hypergeometric random variates—for which suitable algorithms exist—and hence the method is practically feasible.

Alternatively, under the Bayesian paradigm, the missing probability homogeneity between strata may be specified as the model $\mathcal{M}_0$ which assumes that $X_l$ is distributed according to the Binomial law with parameters $n_l$ and $\theta$, for $l = 1, \ldots, L$. In contrast, the model $\mathcal{M}_1$ under the general alternative postulates that $X_l$ be distributed according to the Binomial law with parameters $n_l$ and $\theta_l$, for $l = 1, \ldots, L$. By assuming prior distributions in such a way that $\theta$ is elicited as the absolutely-continuous r.v. $\theta$ defined on $[0, 1]$ with probability density function (p.d.f.) given by $f$, while $(\theta_1, \ldots, \theta_L)$ is elicited as the vector $(_1, \ldots, _L)$ of absolutely-continuous r.v.'s defined on $[0, 1]^L$ with joint p.d.f. given by $f_{1,\ldots,L}$, the Bayes factor is given by

$$
B_{1,0} = \frac{\int_{[0,1]^L} \left\{ \prod_{l=1}^{L} \binom{n_l}{x_l} \theta_l^{x_l}(1 - \theta_l)^{n_l - x_l} \right\} f_{\theta_1,\ldots,\theta_L}(\theta_1, \ldots, \theta_L) d\theta_1 \ldots d\theta_L}{\int_{[0,1]} \left\{ \prod_{l=1}^{L} \binom{n_l}{x_l} \theta^{x_l}(1 - \theta)^{n_l - x_l} \right\} f_\theta(\theta) d\theta}
$$

$$
= \frac{\int_{[0,1]^L} \prod_{l=1}^{L} \theta_l^{x_l}(1 - \theta_l)^{n_l - x_l} f_{\theta_1,\ldots,\theta_L}(\theta_1, \ldots, \theta_L) d\theta_1 \ldots d\theta_L}{\int_{[0,1]} \theta^y (1 - \theta)^{n-y} f_\theta(\theta) d\theta} \ .
$$

If conjugate priors are considered, the r.v. $\theta$ is assumed distributed according to the Beta law with parameters $a$ and $b$, while $(\theta_1, \ldots, \theta_L)$ is the vector of r.v.'s with independent components, in such a way that each $\theta_l$ is distributed according to the Beta law with parameters $a_l$ and

$b_l$. It is worth noting that—in a similar setting—a slightly general hierarchical model is considered by Kass and Raftery [44] (see also [45, p. 190]). Hence, the Bayes factor reduces to

$$B_{1,0} = \frac{B(a, b)}{B(y + a, n - y + b)} \prod_{l=1}^{L} \frac{B(x_l + a_l, n_l - x_l + b_l)}{B(a_l, b_l)} \quad,$$

where—as usual—$B(a, b)$ denotes the Euler's Beta function with parameters $a$ and $b$. In the case of non-informative Uniform priors, i.e. when $a = b = 1$ and $a_l = b_l = 1$ for $l = 1, \ldots, L$, it is apparent that $B_{1,0}$ simplifies to

$$B_{1,0} = \frac{\prod_{l=1}^{L} B(x_l + 1, n_l - x_l + 1)}{B(y + 1, n - y + 1)} \quad.$$

The testing procedures developed above is applied to the data of EXP2 by considering the areas as the strata (see Table 2). At first, by assuming the frequentist paradigm, the null hypothesis $H_0$ of missing proportion homogeneity between strata is considered. The null hypothesis $H_0$ can be rejected since the $P$-value corresponding to the test statistic $R$ was less than $10^{-16}$. Subsequently, by assuming the Bayesian paradigm and non-informative Uniform priors, the Bayes factor is computed. In turn, the missing proportion homogeneity is not likely, since $B_{1,0}$ was less than $10^{-16}$. Thus, the conclusions are as follows. Actually, the adoption of stratified random sampling in EXP2 was a suitable design choice, since the population of articles has a structural partition into areas. However, missing data occurred in the stratified sample, since some reviewers refused to referee the assigned articles. Even if this issue is disturbing, it would be a minor drawback if the items were proportionally missed with respect to the strata. Indeed, in such a case, as showed in Figs 1 and 2, the phenomenon is intrisic in EXP2—owing to the different implementation of EXP2 with respect to EXP1. Generally, if data were missed at random between strata, the effect on the Cohen's kappa estimator could be presumably weak. For a discussion on missing data in the design-based approach, see e.g. the monograph by Little and Rubin [46]. Unfortunately, on the basis of the previous results, we have assessed that the articles are not proportionally missed between the areas, but they are missed according to an unknown random mechanism. As a matter of fact, if the data are missing not at random, corrections are much more difficult and unpredicatable biases could arise [46]. As a consequence, the estimates for EXP2 should be considered very carefully, since in some areas the estimated proportion of missing articles is much more elevate with respect to the other areas: e.g. Area 6 with a missing rate $231/1071 \simeq 21.6\%$ and Area 9 with a missing rate given by $108/739 \simeq 14.6\%$. In addition, these different missing rates occur in the largest strata. Actually, the reasons for which reviewers refused to handle the articles—or to provide the score in the required time—are not known and this issue could introduce a further bias in the results of the assessment.

## 8 Discussion and conclusion

The Italian governmental agency for research evaluation ANVUR conducted two experiments for assessing the degree of agreement between bibliometrics and peer review. They were based on stratified random samples of articles, which were classified by bibliometrics and by informed peer review. Subsequently, concordance measures were computed between the ratings resulting from the two evaluation techniques. The aim of the two experiments was "to validate the dual system of evaluation" [4] adopted in the research assessments. Indeed, in a nutshell, ANVUR used preferentially bibliometric indicators for evaluating articles in the research assessment exercises. When bibliometric rating was inconclusive, ANVUR

commissioned a pair of reviewers to evaluate an article: indeed for these articles peer-review evaluation substituted bibliometrics. Bibliometric and peer reviewer ratings were then summed up for computing the aggregate score of research fields, departments and institutions. The "dual system of evaluation" might have introduced major biases in the results of the research assessments if bibliometrics and peer review generated systematically different scores. A high level of agreement is a necessary condition for the robustness of research assessment results. The two experiments were designed to test the degree of agreement between bibliometrics and peer review at an individual article level.

This paper reconsiders in full the raw data of the two experiments by adopting the same concordance measure—i.e. the weighted Cohen's kappa coefficient—and also the same systems of weights used in EXP1 and EXP2. In view of analyzing the experiments in the appropriate inferential setting, the design-based estimation of the Cohen's kappa coefficient and the corresponding confidence interval were developed and adopted for computing the agreement between bibliometrics and peer review in EXP1 and EXP2. Three suggestions are proposed for defining in a proper way the population Cohen's kappa coefficients to be estimated. In a case, the suggested definition represents the suitable version of the coefficient estimated by ANVUR. The other two definitions are advisable for taking into account the sizes of discarded articles by ANVUR.

As to the agreement between bibliometrics and peer review in EXP1, the point and interval estimates of the considered versions of the weighted Cohen's kappa indicate a concordance degree that can be considered—at most—weak, for the aggregate population and for each scientific area. In EXP2 the degree of agreement between bibliometrics and peer review is generally even lower than in EXP1.

Results for Area 13, i.e. Economics and Statistics, deserve a separate consideration. In EXP1, Cohen's kappa coefficient was estimated to be 54.17%. According to [6], this anomalous high value was possibly due to the modification of the experiment protocol in this area. Indeed, in EXP2—when an identical protocol was adopted for all the areas—the agreement for Area 13 was only slightly larger, but still comparable with the other areas.

Two further points have to be considered. First, the registered lower agreement in EXP2 was arguably due to the adopted systems of ratings, which are based on four categories in EXP1 and on five categories in EXP2. Second, the systems of weights developed by ANVUR tended to boost the value of the weighted Cohen's kappa coefficients with respect to other, more usual, systems of weights (see the S1 Table providing the computations for linear weights). Hence, the estimates indicate that the "real" level of concordance between bibliometrics and peer review is likely to be worse than weak in both EXP1 and EXP2.

The two experiments also investigated the agreement between the two reviewers, when they score each article of the stratified random sample. For EXP1, the correct version of the estimates for the article population indicates that the agreement between the two reviewers tend to be lower than 0.30. A slightly lower concordance level is even obtained for EXP2. In sum, the agreement between pairs of reviewers is weak. In turn, Area 13 represented an exception with the highest level of agreement in both experiments. As previously remarked, in contrast with the other areas, Area 13 adopted a ranking of journals for bibliometric evaluation. When peer reviewers were asked to evaluate a paper, they knew the ranking of journals. Thus, it is possible to conjecture that this very simple information boosted the agreement between reviewers, since they tended to adopt the ranking of journals as a criterion for evaluating articles.

In sum, the two Italian experiments gives concordant evidence that bibliometrics and peer review have less than weak level of agreement at an individual article level. This result is actually consistent with the *Metric Tide* results [11, 47]. Furthermore, they also show that the

agreement between two peer reviewers is in turn very weak. If the agreement between reviewers is interpreted as an estimate of "peer review uncertainty" [2], this uncertainty is of the same order of magnitude of the uncertainty generated by the use of bibliometrics and peer review.

As to EXP2, a further problem arose for the presence of missing values originated by the refusal of some peer reviewers to referee articles of the sample. For EXP2, the results cannot be easily extended even to the population of journal articles submitted to the research assessment.

From the evidence presented in this paper, it is possible to carry out a couple of research policy considerations. The first deals with the Italian research assessments exercises. Results of the experiments cannot be considered at all as validating the use of the dual method of evaluation adopted by ANVUR. At the current state of knowledge, it cannot be excluded that the use of the dual method introduced uncontrollable major biases in the final results of the assessments. Indeed, bibliometrics and peer review show a weak agreement. In particular, the evidence drawn from data in the official research reports [12, 17] shows that peer reviewers' scores were on average lower than bibliometric ones. Unbiased results at an aggregate level would be produced solely if the distribution of articles evaluated by the two methods was homogenous for the various units of assessment (research field, research area, departments and universities). Official reports show that the distribution was not homogenous. The distributions per research areas of the articles with an inconclusive bibliometric score and consequently evaluated by peer review varied from 0.9% to 26.5% in VQR1 (source: [12, Table 3.5]), and from 0.1% to 19.2% in VQR2 (source: [17, Table 3.5]). Therefore, the aggregate results for research fields, departments and universities might be affected by the proportion of research outputs evaluated by the two different techniques: the higher the proportion of research outputs evaluated by peer review, the lower the aggregate score. From publicly available data, it is possible to show that the average score at the research area level has—rather generally—a negative association with the percentage of papers evaluated by peer review. This issue actually holds for VQR1 and VQR2, as shown in the S5–S8 Figs (data available as S2 File). These considerations do not permit to exclude that the results of two Italian research assessments are biased. As a consequence, it is questionable their use for policy purposes and funding distribution.

Generally, the lesson from the two Italian experiments is that the use of a dual method of evaluation in the same research assessment exercise should be at least considered with extreme caution. A low agreement between bibliometrics and peer review at the level of individual article indicates that metrics should not replace peer review at the level of individual article. The use of the dual methods for reducing costs of evaluation, might dramatically worsen the quality of information obtained in a research assessment exercise.

## Supporting information

**S1 Fig. Joint distribution of peer review and bibliometrics evaluations for EXP1.** Count overlapping points (proportion). All research areas.
(TIF)

**S2 Fig. Joint distribution of peer review and bibliometrics evaluations for EXP1.** Count overlapping points (proportion). Separate plot for each research area.
(TIF)

**S3 Fig. Joint distribution of peer review and bibliometrics evaluations for EXP2.** Count overlapping points (proportion). All research areas.
(TIF)

**S4 Fig. Joint distribution of peer review and bibliometrics evaluations for EXP2.** Count overlapping points (proportion). Separate plot for each research area.
(TIF)

**S5 Fig. Average score and percentage of articles evaluated by peer review for research fields in VQR1.** All areas.
(TIF)

**S6 Fig. Regression lines for average score and percentage of articles evaluated by peer review for different research areas.** VQR1.
(TIF)

**S7 Fig. Average score and percentage of articles evaluated by peer review for research fields in VQR2.** All areas.
(TIF)

**S8 Fig. Regression lines for average score and percentage of articles evaluated by peer review for different research areas.** VQR2.
(TIF)

**S1 File. Raw anonymized data of the experiments EXP1 and EXP2.**
(XLSX)

**S2 File. Average score and proportion of peer review evaluations at a scientific field level.** Data from the two Italian research assessment exercises VQR1 and VQR2.
(XLSX)

**S1 Table. Cohen's kappa estimates with linear weights for EXP1 and EXP2.** 4 Tables.
(PDF)

## Author Contributions

**Conceptualization:** Alberto Baccini, Lucio Barabesi, Giuseppe De Nicolao.

**Data curation:** Alberto Baccini, Giuseppe De Nicolao.

**Formal analysis:** Alberto Baccini, Lucio Barabesi.

**Funding acquisition:** Alberto Baccini.

**Investigation:** Alberto Baccini, Giuseppe De Nicolao.

**Methodology:** Alberto Baccini, Lucio Barabesi, Giuseppe De Nicolao.

**Software:** Lucio Barabesi.

**Validation:** Alberto Baccini, Lucio Barabesi, Giuseppe De Nicolao.

**Visualization:** Alberto Baccini, Lucio Barabesi.

**Writing – original draft:** Alberto Baccini, Lucio Barabesi, Giuseppe De Nicolao.

**Writing – review & editing:** Alberto Baccini, Lucio Barabesi, Giuseppe De Nicolao.

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
