## [Decision Letter · Decision Letter 0]

7 Sep 2020

PONE-D-20-16421

On the agreement between bibliometrics and peer review: evidence from the Italian research assessment exercises

PLOS ONE

Dear Dr. Baccini,

Thank you for submitting your manuscript to PLOS ONE. After careful consideration, we feel that it has merit but does not fully meet PLOS ONE’s publication criteria as it currently stands. Therefore, we invite you to submit a revised version of the manuscript that addresses the points raised during the review process.

We look forward to receiving your revised manuscript.

Kind regards,

Filippo Radicchi, Ph.D.

Academic Editor

PLOS ONE

Journal Requirements:

Reviewers' comments:

Reviewer's Responses to Questions

**Comments to the Author**

1. Is the manuscript technically sound, and do the data support the conclusions?

Reviewer #1: Yes

Reviewer #2: No

2. Has the statistical analysis been performed appropriately and rigorously? 

Reviewer #1: Yes

Reviewer #2: Yes

3. Have the authors made all data underlying the findings in their manuscript fully available?

Reviewer #1: Yes

Reviewer #2: Yes

4. Is the manuscript presented in an intelligible fashion and written in standard English?

Reviewer #1: Yes

Reviewer #2: Yes

5. Review Comments to the Author

Reviewer #1: The authors aimed to replicate an analysis conducted by an Italian government agency that showed a fundamental agreement between bibliometrics and peer reviews in an Italian research evaluation exercise.

The analysis was based on samples of journal articles that were evaluated by peer reviews and bibliometrics, and their agreement was measured by the weighted Cohen's kappa coefficient and its variant, with considerations to the small sample size and disciplinary differences.

The agreements shown by the agency are often quoted as evidence to support the adoption of the Italian's dual research evaluation systems.

The authors pointed out the biases which potentially misled the analysis by the agency arising from the experimental design, the sampling method, and the treatment to missing values.

When correcting the biases, the authors showed a weak agreement between the two evaluation systems, a finding that contradicts that in the official report from the agency.

The authors deliver the purpose, method, and result very clearly. Their results have profound implications for the use of bibliometrics in research assessment exercises that may be broadly applied to research exercise in general. This is a very well written paper and I recommend it for publications in the PLOS ONE.

I leave some minor points in the following:

- p. 9 line 284 "(downloadable also from https://doi.org/10.5281/zenodo.3727460)"

The page is not reachable.

- p. 10 Eq. (1)

Missing comma at the end of the equation (the one between 'while' and 'and').

- p. 13

To be mathematically precise with the definition, the irreducible numbers should be expressed as a fractional number, e.g., rewrite 0.67 as 2/3.

- p. 18 Section 7

The authors demonstrated that the missing peer ratings are distributed heterogeneously, raising caution in interpreting the results for the EXP2. While two statistical tests support this claim, I would like to see a short discussion regarding the actual impact of the missing values. In fact, it is possible that the actual impact is weak and thus negligible even if the heterogeneity is statistically significant.

Reviewer #2: The manuscript assesses the concordance between bibliometric indicators and peer review by developing variants of Cohen's kappa coefficient that are suited in the design-based setting. The results indicate a weak agreement between bibliometrics and peer review.

Some justifications are needed to explain why using kappa estimators. Why not using many of the regression frameworks that have already been developed in the context of evaluating the agreement of peer review of grant proposals and their outcomes?

With respect to kappa estimation, the manuscript lacks proper baseline values against which one can compare the observed values. Is a Cohen’s kappa coefficient 0.3 high or low to allow one to conclude that there is a weak agreement? This measure of course considers the probability of agreement by chance, which assumes that the raters observe each category randomly. But does this assumption reflect the experiment setting described in the manuscript?

It is not clear how the composite bibliometric indicators which combine citation count and impact factor are developed and binned into different categories, as kappa coefficient requires discrete values rather than continuous variables.

It is concluded that the dual system “might introduce unknown biases.” But could it also be the case that both human ratings and bibliometric measures introduce biases? There are studies showing that there are systematic biases when experts evaluating grant proposals, and many bibliometric studies have revealed biases of bibliometric indicators.

Some clarifications are needed to justify why a design-based Cohen’s kappa coefficient is developed. How is it different from the ordinary kappa coefficient?

There have been some debates about the terms of “hard science” and “soft science”. I’d suggest voiding using “hard science” and instead to list the areas explicitly.

It may be useful to summarize the existing literature presented in Section 2 in a table.

It would also be really helpful for the readers to understand the two experiments if there were a flowchart to illustrate the process described in Section 3.

Page 6: It becomes clear later in the manuscript what “IR” is. And what is “IPR”?

6. PLOS authors have the option to publish the peer review history of their article (what does this mean?). If published, this will include your full peer review and any attached files.

Reviewer #1: No

Reviewer #2: No

---

## [Author Response · Author response to Decision Letter 0]

2 Oct 2020

Reply to Referees

1 Reviewer #1

1.1 Main comments.

The authors aimed to replicate an analysis conducted by an Italian government agency that showed a fundamental agreement between bibliometrics and peer reviews in an Italian research evaluation exercise. The analysis was based on samples of journal articles that were evaluated by peer reviews and bibliometrics, and their agreement was measured by the weighted Cohen’s kappa coefficient and its variant, with considerations to the small sample size and disciplinary differences. The agreements shown by the agency are often quoted as evidence to support the adoption of the Italian’s dual research evaluation systems. The authors pointed out the biases which potentially misled the analysis by the agency arising from the experimental design, the sampling method, and the treatment to missing values. When correcting the biases, the authors showed a weak agreement between the two evaluation systems, a finding that

contradicts that in the official report from the agency. The authors deliver the purpose, method, and result very clearly. Their results have profound implications for the use of bibliometrics in research assessment exercises that may be broadly applied to research exercise in general. This is a very well written paper and I recommend it for publications in the PLOS ONE.

Reply.

Thank you very much for your appreciation.

1.2 Minor comments.

- p. 9 line 284 “(downloadable also from https://doi.org/10.5281/zenodo.3727460)”. The page is not reachable.

Reply. Fixed. The page is reachable.

- p. 10 Eq. (1) Missing comma at the end of the equation (the one between ’while’

and ’and’).

Reply. Done.

- p. 13 To be mathematically precise with the definition, the irreducible numbers should be expressed as a fractional number, e.g., rewrite 0.67 as 2/3.

Reply. Done.

- p. 18 Section 7. The authors demonstrated that the missing peer ratings are distributed heterogeneously, raising caution in interpreting the results for the EXP2. While two statistical tests support this claim, I would like to see a short discussion regarding the actual impact of the missing values. In fact, it is possible that the actual impact is weak and thus negligible even if the heterogeneity is statistically significant.

Reply. First, it is worth noting that the proportion of missing data in EXP2 is not negligible, since it equals 503/6041 ' 8.3% in the sample. In any case, there exist appropriate techniques for dealing with data involving such a missing rate (see e.g. the recent edition of the authoritative monograph by Little, R.J.A. and Rubin, D.B.

(2020) Statistical Analysis with Missing Data, 3rd ed., Wiley, Hoboken). However, a great deal depends on the reasons for dropout. If the data are missing at random, then a suitable approach - such as imputation - may result in unbiased, or at least much less biased, estimation. In the present case, if data were missed at random between strata, the effect on the Cohen’s kappa estimator could be arguably weak. Unfortunately, if the data are missing not at random, corrections are much more difficult and unpredicatable biases could arise. In the present case, in some areas (e.g.

Area 6 with a missing rate 231/1071 ' 21.6% and Area 9 with a missing rate given by

108/739 ' 14.6%) the estimated proportion of missing articles is much more elevate with respect to the other areas. In addition, these different missing rates occur in the largest strata. For these reasons, the estimates for EXP2 should be considered carefully. Hence, we have add further comments to the discussion at the end of Section 7.

Thank you very much for your careful reading of the paper and for your constructive comments. We hope we have correctly addressed your concerns and comments.

Best regards.

Alberto Baccini, Lucio Barabesi, Giuseppe De Nicolao

2 Reviewer #2

2.1 Main comments

The manuscript assesses the concordance between bibliometric indicators and peer review by developing variants of Cohen’s kappa coefficient that are suited in the

design-based setting. The results indicate a weak agreement between bibliometrics and peer review.

- 1 - Some justifications are needed to explain why using kappa estimators. Why not using many of the regression frameworks that have already been developed in the context of evaluating the agreement of peer review of grant proposals and their outcomes?

Reply. We agree with the reviewer that ANVUR’s choice of weighted Cohen’s kappa for evaluating agreement is not optimal. However, since our starting point was to reproduce ANVUR’s computations, we was obliged to adopt weighted Cohen’s kappa. In Section 4 of the paper we have more explicitly highlighted this point.

- 2 - With respect to kappa estimation, the manuscript lacks proper baseline values against which one can compare the observed values. Is a Cohen’s kappa coefficient 0.3 high or low to allow one to conclude that there is a weak agreement? This measure of course considers the probability of agreement by chance, which assumes that the raters observe each category randomly. But does this assumption reflect the experiment setting described in the manuscript?

Reply. We have discussed more explicitly the guidelines for interpreting Cohen’s kappa values. We have introduced the available guidelines on the first part of section

4, and we have discussed the adopted baseline on section 6. The baseline for interpreting these values is provided in Table 13.6 by Fagerland et al., Statistical Analysis of Contingency Tables, 2017. According to this guideline, a value of the simple Cohen’s kappa less than or equal to 0.20 is considered as a “poor” concordance and a value in the interval (0.20, 0.40] is considered as a “weak” concordance; values in the intervals (0.40, 0.60] and (0.60, 1.00] are considered respectively as indicating a “moderate” and a “very good” concordance. However, it should be remarked that these considerations are carried out for the simple Cohen’s kappa. Hence, in the present

case, the small values of the weighted Cohen’s kappa coefficients can be interpreted as indicating a concordance even worse than weak.

- 3 - It is not clear how the composite bibliometric indicators which combine citation count and impact factor are developed and binned into different categories, as kappa coefficient requires discrete values rather than continuous variables.

Reply. In section 3 of the paper we have clarified that the algorithms which combine citation count and impact factor result in categorical scores. We have highlighted the evaluation categories adopted in EXP1 and EXP2. We have referenced the available papers that present the algorithms and their drawbacks. The algorithms have been developed by ANVUR. The raw data are not publicly available. We have received anonymous data where each journal article is assigned to a category, but it is impossible to appreciate or to replicate the functioning of the algorithms.

- 4 - It is concluded that the dual system “might introduce unknown biases.” But could it also be the case that both human ratings and bibliometric measures introduce biases? There are studies showing that there are systematic biases when experts evaluating grant proposals, and many bibliometric studies have revealed biases of bibliometric indicators.

Reply. The reviewer is right. We had really left the expression “might introduce unknown biases” in the summary but the conclusion of the article sounded slightly different. We have therefore reformulated the point more precisely by writing: “the use of the dual system in a research assessment might worsen the quality of information compared to the adoption of peer review only or bibliometrics only”. The basic intuition is that with peer review or bibliometrics only, we have only one source of bias. The adoption of the dual system simply mix the two biases in an unknowable way.

- 5 - Some clarifications are needed to justify why a design-based Cohen’s kappa coefficient is developed. How is it different from the ordinary kappa coefficient?

Reply. - We were forced to introduce a suitable definition of the population Cohen’s kappa coefficient and its estimation in the design-based approach since it seems that this issue was not previously considered in literature – at the best of our knowledge. Actually, ANVUR has analyzed data arising from a sampling design, even if has considered a model-based approach in order to carry out inference on the Cohen’s kappa. In order to be more explicit, we have introduced a detailed discussion of the model-based approach in Section 5 before introducing our proposal in the design-based approach.

6 - There have been some debates about the terms of “hard science” and “soft science”. I’d suggest voiding using “hard science” and instead to list the areas explicitly.

Reply. - Done. We are using now a generic “science, technology, engineering and mathematics in the abstract and in the introductory section for syntetic purposes. But in the discussion we use the names of the areas explicitly.

7 - It may be useful to summarize the existing literature presented in Section 2 in a table.

Reply. The literature review is discussed shortly in Section 2. Since we have added two new figures (see the next reply), and the paper has also many supplementary materials, we prefer not to strech it by introducing also a new table.

- 8 - It would also be really helpful for the readers to understand the two experiments if there were a flowchart to illustrate the process described in Section 3.

Reply. Many thanks for the suggestion. We have introduced two flowcharts (Fig 1 and Fig 2) describing the two experiments.

- 9 - Page 6: It becomes clear later in the manuscript what “IR” is. And what is

“IPR”?

Reply. IPR is the acronym used by ANVUR for Informed Peer Review. IPR indicates that reviewers were asked to evaluate a submitted research item by being provided with its complete metadata and, if available, with its bibliometric indicators. We have clearly stated the point in 

footnote 7. We have also clarified the IPR process and its outcome by revising Section 3.

Thank you very much for your careful reading of the paper and for your insightful constructive comments. We hope we have correctly addressed your concerns and comments.

Best regards,

Alberto Baccini, Lucio Barabesi, Giuseppe De Nicolao

3 Editor

Dear Editor,

We are very grateful for handling our work and for providing a prompt decision on it. In this new manuscript we have fully addressed all the concerns and comments of the reviewers. As a result, we believe that our work has improved. We hope that the new submission could meet PLOS ONE’s standards.

Best regards,

Alberto Baccini, Lucio Barabesi, Giuseppe De Nicolao

---

## [Decision Letter · Decision Letter 1]

4 Nov 2020

On the agreement between bibliometrics and peer review: evidence from the Italian research assessment exercises

PONE-D-20-16421R1

Dear Dr. Baccini,

We’re pleased to inform you that your manuscript has been judged scientifically suitable for publication and will be formally accepted for publication once it meets all outstanding technical requirements.

Kind regards,

Filippo Radicchi, Ph.D.

Academic Editor

PLOS ONE

Additional Editor Comments (optional):

Reviewers' comments:

Reviewer's Responses to Questions

**Comments to the Author**

1. If the authors have adequately addressed your comments raised in a previous round of review and you feel that this manuscript is now acceptable for publication, you may indicate that here to bypass the “Comments to the Author” section, enter your conflict of interest statement in the “Confidential to Editor” section, and submit your "Accept" recommendation.

Reviewer #1: All comments have been addressed

Reviewer #2: All comments have been addressed

2. Is the manuscript technically sound, and do the data support the conclusions?

Reviewer #1: Yes

Reviewer #2: Yes

3. Has the statistical analysis been performed appropriately and rigorously? 

Reviewer #1: Yes

Reviewer #2: Yes

4. Have the authors made all data underlying the findings in their manuscript fully available?

Reviewer #1: Yes

Reviewer #2: Yes

5. Is the manuscript presented in an intelligible fashion and written in standard English?

Reviewer #1: Yes

Reviewer #2: Yes

6. Review Comments to the Author

Reviewer #1: The authors addressed all of typos and my concerns. The manuscript has been much improved and thus recommend for the publication.

Reviewer #2: (No Response)

7. PLOS authors have the option to publish the peer review history of their article (what does this mean?). If published, this will include your full peer review and any attached files.

Reviewer #1: No

Reviewer #2: No

---

## [Editor Report · Acceptance letter]

9 Nov 2020

PONE-D-20-16421R1 

On the agreement between bibliometrics and peer review: evidence from the Italian research assessment exercises 

Dear Dr. Baccini:

I'm pleased to inform you that your manuscript has been deemed suitable for publication in PLOS ONE. Congratulations! Your manuscript is now with our production department. 

Kind regards, 

on behalf of

Dr. Filippo Radicchi 

Academic Editor

PLOS ONE